# Estimation of Degree of Sea Ice Ridging Based on Dual-Polarized C-band SAR Data

Alexandru Gegiuc, Markku Similä, Juha Karvonen, Mikko Lensu, Marko Mäkynen, and Jouni Vainio

Finnish Meteorological Institute(FMI), Marine Research, Erik Palménin aukio 1, 00560 Helsinki, Finland

*Correspondence to:* Alexandru Gegiuc (alexandru.gegiuc@fmi.fi)

**Abstract.**

For ship navigation in the Baltic Sea ice, parameters such as ice edge, ice concentration, ice thickness and degree of ridging are usually reported daily in manually prepared ice charts. These charts provide icebreakers essential information for route optimization and fuel calculations. However, manual ice charting requires long analysis times and detailed analysis of large areas (e.g. Arctic Ocean) is not feasible. Here, we propose a method for automatic estimation of the degree of ice ridging in the Baltic Sea region, based on RADARSAT-2 C-band dual-polarized (HH/HV channels) SAR texture features and sea ice concentration information extracted from the Finnish ice charts. The SAR images were first segmented and then several texture features were extracted for each segment. Using the Random Forest classification method, we classified them into four classes of ridging intensity and compared them to the reference data extracted from the digitized ice charts. The overall agreement between the ice chart based degree of ice ridging and the automated results varied monthly, being 83 %, 63 % and 81 % in January, February and March 2013, respectively. The correspondence between the degree of ice ridging reported in the ice charts and the actual ridge density was validated with data collected during a field campaign in March 2011. In principle the method can be applicable to the seasonal sea ice regime in the Arctic Ocean.

## 1 Introduction

Navigation in sea ice is hampered by rapid changes in the sea ice conditions. Thus, it is essential for winter time shipping and off-shore operations to get reliable and up-to-date information on the prevailing ice conditions. The most important sea ice parameters are the location of the ice edge, ice types, ice thickness, concentration and the amount of ridged ice. Without detailed sea ice information, navigating through heavily ridged sea ice is difficult or even impossible.

The Baltic Sea is a semi-enclosed brackish sea water basin in the Northern Europe. The ice cover in the Baltic Sea usually begins to form in November, and has its largest extent between January and March (Seinä and Peltola, 1991). The normal melting season starts in April, and the ice melts completely by the beginning of June. The maximum annual ice extent ranges from 12% to 100% of the whole Baltic Sea area with an average of 40% (Seinä and Palosuo, 1996). During the last decades a clear decreasing trend can be seen in the maximum ice extent although the trend has not yet been a subject for a detailed investigation. The sea ice edge is a parameter of sea ice defining the boundary between the open-water and the sea ice area. Here the ice concentration areas higher than 25 % belong to sea ice, and below to open-water. The sea ice in the Baltic Sea

can be divided into fast ice and drift ice. Fast ice appears in the coastal and archipelago areas. Drift ice has a dynamic nature due to forcing by winds and currents, which results in an uneven broken ice field with distinct floes, leads and cracks, brash ice barriers, rafted ice and ice ridges. The upper limit for thermodynamically grown ice in the drift ice zone is 70 cm or less during most winters (Palosuo et al., 1982) while the keel depth of ice ridges is typically 5 to 15 m (Leppäranta and Hakala, 1992). The salinity of the Baltic Sea ice is typically only from 0.2 to 2 ‰ depending on the location, time, and weather history (Hallikainen, 1992). The low salinity level affects the radar signal response from satellite imagery, resulting in more volume and less surface scattering of the incident signal.

Synthetic Aperture Radar (SAR) satellites such as RADARSAT-2 (RS-2) and Sentinel-1 (S-1) play a major role in operationally monitoring the ice conditions in the Baltic Sea. SAR imaging is practically independent of the atmosphere conditions (e.g. cloud cover) and solar illumination and therefore suitable for operative sea ice monitoring. However, as backscatter information in the SAR imagery cannot easily be linked to the different ice types, the expertise of an trained ice analyst is usually required.

In the Baltic Sea, daily ice charts prepared by the Finnish Ice Service (FIS) analysts provide a daily source of information upon the ice conditions. The charts divide the ice cover into polygons for which ice types and properties are assigned. The analysis is based on visual interpretation of the SAR imagery as the principal source of ice information. Currently, RS-2 and S-1 C-band SAR imagery with a wide coverage (e.g. RS-2 ScanSAR Wide Mode with 500 by 500 km image size) are used. The SAR imagery is complemented by visible and thermal infrared imagery from e.g. Moderate Resolution Imaging Spectro-radiometer (MODIS), in-situ observations, sea ice information messages from icebreakers, and data from sea ice models. The ice chart polygons defined by the ice analysts represent ice areas with similar ice characteristics. Parameters assigned for each polygon are ice concentration, average level ice thickness, maximum and minimum level ice thickness, and the the degree of ice ridging (DIR) which is a numeral classifying the ice into five categories, as explained in the section 2.3. The FIS ice analysts estimate the DIR values mainly using SAR imagery, and with additional information on the ice drift based on successive SAR images and results of sea ice models. The main criteria for the visual DIR estimation from the SAR imagery are the SAR backscattering and its visible patterns (SAR texture) (see Fig. 1).

Typically, the ice situation changes little from one day to the next. Hence, when drawing a new ice chart, the ice analysts are using the latest chart as the basis for the new one, and only adjusting the polygon contours and their assigned DIR values to match the new ice situation. This procedure speeds up the process of ice charting but may also introduce a bias if old polygons are used. The quality of the displayed SAR features of sea ice (e.g. magnitude of contrast/intensity, amount of radar noise), the analyst's experience and their style of drawing (more detailed or less detailed) can contribute further to inconsistencies in the finalized ice chart.

In this paper we propose a method to automatize the DIR estimation process based on RS-2 dual-polarized (HH/HV) SAR data acquired under cold conditions and using the FIS ice charts as reference data. The results are then evaluated together with the ice analysts. We don't expect a perfect match between the automatic chart and the manual one. The polygons in the manual charts suppress variation for the small-scale features and merge them into one DIR category. Here we aim to produce a more detailed DIR chart, which closely follows the SAR texture features of sea ice ridges, edges, cracks and leads. This would allow

the icebreakers and non-icebreaker vessels to benefit from it in advance route planning and optimization, by taking advantage of the sea ice passages within ridged ice areas. Manual ice charting should also benefit from a more detailed and automated DIR map which can serve as basis layer for the final ice chart.

In section 2.3 we will describe the different DIR categories used by FIS. As a tool in the DIR classification we use the
Random Forest (RF) algorithm which will be explained in detail in Section 3.3. Using the automated classification procedure we target an efficient exploitation of SAR data and, by means of increased spatial and temporal resolutions, an improved quality (pixel level accuracy and consistency between different analysts) of the ice charts.

## 2   Data Sets and Processing

Our study area in the Baltic Sea is northward from the latitude of 61° N, covering the entire Bay of Bothnia and most of the
Sea of Bothnia. The time period is the ice season 2012–2013. The most severe ice conditions in 2012–2013 in the Baltic Sea occurred in our study area. The ice season 2012–2013 was average but the turning point of the winter was late. The weather began to cool during the first week of January and the ice extent increased. In the last week of January the strong winds moved the ice fields and the mild weather melted ice. In the beginning of February the weather remained similar – at night new ice was formed and then broken by winds during the day. Towards the end of February the weather cooled down and new ice was
formed also in the Gulf of Finland. In the beginning of March cold arctic air started to flow to Scandinavia and the extent of ice began to grow. The whole March was extremely cold. The 15[th] of March ice extent reached 177 000 km$^2$, which was the maximum of the season. From then on, the cold nights formed new ice but sunny days melted it, and the ice extent did not increase further.

### 2.1   $\sigma^o$ of the Baltic Sea ice

In the following we discuss how in general the C-band backscattering coefficient ($\sigma^o$) of the Baltic Sea ice is related to the sea ice properties, and especially sea ice ridging. Under cold weather conditions when the snow cover on sea ice is dry the ice surface scattering has been observed to be the dominant component in the total co-polarized $\sigma^o$ at incidence angles below 45° (Carlström and Ulander, 1993; Dierking et al., 1999). If the ice surface is very smooth and salinity < 0.5 psu, which typically is the case for level fast ice in the Baltic Sea, then the backscattering from ice-water interface and ice volume is significant. The
surface backscattering from level ice is controlled by the statistics of the small-scale roughness as well as the salinity of the ice surface. If sea ice is ridged, the large-scale surface roughness alters the geometry of the surface and, hence, also modifies $\sigma^o$. Empirical measurements of the Baltic Sea ice C-band $\sigma^o$ have shown that the variation in the large-scale surface roughness mostly modulates $\sigma^o$ and image texture although changes in the small-scale roughness are also significant (Carlström and Ulander, 1993; Dierking et al., 1999; Mäkynen and Hallikainen, 2004).

The $\sigma^o$ contrast between level ice and ridged ice is on average larger at C-band cross-polarization than at co-polarization (Mäkynen and Hallikainen, 2004). The standard deviation of $\sigma^o$ was observed to be larger for ridged ice types (mixtures of level ice, ice ridges, rubble) than for level ice types and brash ice in Mäkynen and Hallikainen (2004). The C-band $\sigma^o$ is not

directly related to the sea ice thickness, but at least in the Baltic Sea it is possible to estimate the thickness of ridged ice under dry snow conditions through a statistical relationship between ice freeboard, level ice thickness and $\sigma^o$ (Similä et al., 2010). The variance of the mean freeboard, i.e. large scale surface roughness, increases with increasing average freeboard, and as the surface roughness increases $\sigma^o$ also typically increases. In general, these previous studies on sea ice $\sigma^o$ signatures show that there is a relation between C-band $\sigma^o$ and DIR, but further studies are needed to better quantify this relation.

Next, different approaches for SAR based sea ice classification are briefly reviewed. Many SAR imagery based sea ice classification systems just perform classification to open water and different ice types, such as new ice, first-year-ice, multiyear-ice, but DIR is not explicitly estimated in more detail. Classification schemes utilizing $\sigma^o$ and SAR texture have been presented e.g. in Soh et al. (2004); Sandven et al. (2012); Barber and LeDrew (1991) and Clausi (2001). Classification of ice types based on single-polarization C-band SAR backscattering has been studied e.g. in Karvonen (2004) and Shokr (2009). Sea ice SAR classification using the world meteorological organization (WMO) ice categories (stage of development) (WMO, 2010) has been studied e.g. in Clausi (2001); Deng and Clausi (2005); Maillard et al. (2005); Yu and Clausi (2007); Clausi et al. (2010); Ochilov and Clausi (2012). These approaches are based on the SAR segmentation and different SAR features, including texture ones. Some of the methods also combine the ice analyst analysis and an automated analysis. A system capable of a semi-automated segmentation and enhanced classification with a digitized ice chart as an input is presented in Clausi et al. (2010). It is noted that the ice categories in these studies do not either explicitly or uniquely include DIR classification.

## 2.2 RADARSAT-2 SAR imagery

The SAR imagery used in this study is RADARSAT-2 ScanSAR Wide (SCWA) dual-polarized imagery with the HH/HV polarization combination. The nominal size of an RS-2 SCWA image is around 500 by 500 km, and the pixel size is 50 m. The spatial resolution is around 73-163 m by 100 m (range by azimuth). The incidence angle ($\theta_0$) varies from 20 to 49 degrees. The equivalent number of looks (ENL) is larger than six. The nominal noise floor equivalent $\sigma^o$ at both HH- and HV-polarization varies along the across-track direction as -28.5 $\pm$ 2.5 dB and the absolute accuracy of $\sigma^o$ is better than 1 dB (MDA, 2014).

The acquired RS-2 SAR imagery covered the whole Baltic Sea. The number of SAR images used in the daily SAR mosaic over the test area varied from one to three SAR frames per day from January to March in 2013. On some days the SAR mosaic was updated twice. We selected from these SAR mosaics the training and the test data using the rule that the time gap between a training and a test mosaic must be at least three days to avoid situations where a same SAR scene would appear both in the test and the training data set. Hence the training data consisted of 5 mosaics from January, 4 mosaics from February and 4 mosaics from March. The test data consisted of 4 mosaics from January, 6 mosaics from February and 3 mosaics from March. Some mosaics were not used due to the time constraint. The monthly training and test data refers to the mosaics gathered during the same month. We selected these three months for the development and the test work because then the SAR images were mostly acquired under dry snow conditions. Hence, the dominant backscattering source was the sea ice surface and we could expect a statistical relationship between $\sigma^o$ and DIR as reported in Carlström and Ulander (1995); Dierking et al. (1999); Similä et al. (2001); Mäkynen and Hallikainen (2004); Similä et al. (2010).

The preprocessing of the RS-2 SCWA images consisted of calibration (calculation of $\sigma^o_{HH}$ and $\sigma^o_{HV}$ ), georectification, calculation of the incidence angle $\theta_0$, and land masking. First the data were rectified into the Mercator projection with 100 m pixel size. This georectification is compatible with the FIS ice charts and the navigation system of the Finnish and Swedish icebreakers. In this Mercator projection the reference latitude is 61°40' N.

As the SAR $\sigma^o$ is dependent on $\theta_0$, an incidence angle correction is necessary before the classification of the SAR images with wide $\theta_0$ range, such as RS-2 SCWA images. For the HH-polarization images, an incidence angle correction method described in Mäkynen et al. (2002) was applied. This incidence angle correction maps the $\sigma^o$ values using a linear dependence for the $\sigma^o$ in dB-scale to a predefined $\theta_0$ value $\theta_0^R$. In this case, the fixed $\theta_0^R$ of 30° was used.

At the HV-polarization the SAR $\sigma^o$ values are close to the instrument noise floor (around -28.**5** dB for RS-2 SCWA mode). The noise floor modulates the (low) HV channel signal leading to clearly visible stripes (artifacts) in the HV imagery. These stripes complicate both the visual and automated interpretation of the SAR imagery. The HV channel incidence angle dependence and varying noise floor in the range direction are corrected based on a statistical incidence angle dependence computed for a large number (65) of RS-2 ScanSAR wide images. Then the incidence angle range is quantized into N bins of 0.01 degrees $(\theta_1, .., \theta_N)$ covering the whole incidence angle range. We average the $\sigma^o$ values from all the 65 images for each $\theta_i$. We denote this average as $\bar{\sigma}^o(\theta)$. Together these average values create a $\sigma^o$ curve Z($\theta$) as a function of the incidence angle over the whole incidence angle range. We consider that each $\bar{\sigma}^o(\theta)$ consists of the sum of the average $\sigma^o$, the associated noise floor and the $\sigma^o$ decay as a function of the incidence angle. As we average a large number of values representing different targets for each theta bin, we assume the $\sigma^o$ average to be similar for each bin (i.e. constant over the whole incidence angle range), and then Z($\theta$) also presents a sum of the constant value, the noise floor as a function of theta and $\sigma^o$ decay as a function of theta. To minimize the effect of the constant value on the correction we then take the mean of all Z($\theta$) values over the whole range of theta. We denote this mean as $\bar{Z}$. Then the whole Z($\theta$) curve is made a zero mean signal by subtracting $\bar{Z}$ from Z($\theta$), this zero mean function is denoted by $\bar{Z}(\theta)$. The incidence angle corrected backscattering coefficient $\sigma^o_{HV_{corrected}}(r,c)$ is now obtained from each calibrated pixel at $\sigma^o(r,c)$ value: $\sigma^o_{HV_{corrected}}(r,c) = \sigma^o(r,c) - \bar{Z}(\theta(r,c))$, where $(r,c)$ are the row and column coordinates of the image grid and $\theta(r,c)$ is the corresponding incidence angle. This correction was proposed in (Karvonen, 2015).

The equivalent number of looks (ENL), noise equivalent $\sigma^o$ and autocorrelation between neighboring pixels in the rectified images were studied using homogeneous areas of size 3.1x3.1 km visually selected from the images over open water areas with a weak texture. The ENL was around 9.5 for the whole $\theta_o$ range. Thus, the radiometric resolution was around 1.2 dB and the standard deviation (std) of fading was 1.4 dB. The autocorrelation coefficient between the adjacent 100 m pixels was on the average only 0.18. The land masking was based on the GSHHG (Global Self-Consistent Hierarchical, High-resolution Geography database from the National Oceanic and Atmospheric Administration, NOAA) coastline data (Wessel and Smith, 1996).

Next, the SAR images were segmented, and the segmentwise features were calculated at the resolution of 100 m, for details see Sections 3.1 and 3.2. Due to the large size of the SAR images and also the feature images they were downsampled into 500 m resolution. Finally, the daily SAR image and features mosaics were constructed by overlaying all of the SAR data

available for each day, i.e. the latest data is shown in the mosaic. The study area was typically fully covered by RS-2 SAR imagery every 1 to 2 days.

## 2.3 Ice charts and degree of ridging

Our reference data set consists of the daily FIS manual ice charts over the Baltic Sea. In the ice charts the degree of ice
ridging (DIR) is used to classify sea ice in a way that is relevant for the difficulty of navigation. DIR is manually assigned as a qualitative numeral, ranging from 1 to 6, to each ice chart polygon. The six DIR categories used in the operational ice charting in the Baltic Sea, relate to the ridge density variation in an area. The categories are visually identified through changes both in the $\sigma^o$ response as well as in textural characteristics in SAR imagery. Their interpretation is validated through field measurements provided by the several operating icebreakers. The task of assigning a DIR value to each ice chart polygon is
a complex process requiring a good understanding of the history of the current winter season, i.e. monitoring of changes in the pack ice zone and utilizing the continuous reports on ice conditions provided by the icebreakers. In our study, we only define four DIR categories by combining the brash ice barriers (WMO, 2010) and the heavily ridged ice category. The brash ice barriers covered a very small fraction of the sea ice area, so that they couldn't have been treated as a distinct category for classification. The very heavily ridged ice field category was not present in our dataset. The four DIR categories used in this
study are defined as follows:

The level-ice area is indicated as DIR 1 which usually looks homogenous and smooth in the SAR imagery with low $\sigma^o$ response. This category includes also slightly rafted ice. The slightly ridged ice category including heavily rafted ice areas is marked as DIR 2. The ridged and heavily ridged ice areas corresponding to the DIR 3 and 4, respectively, are recognized by the changes in surface roughness at a larger scale resulting in higher $\sigma^o$ values. As their formation depends on ice pressure,
knowledge of the earlier ice and weather conditions is required. The DIR 4 category in our data set included the few occurrences of brash ice barriers.

By visual inspection of the RS-2 and S-1 SAR wide swath imagery with a spatial resolution of approximately 100 m it is not possible to describe the ridging intensity quantitatively. However, it is feasible to assign categories of ridged ice for extended areas for which the actual ridging intensities differ. For justification of the meaningfulness of the areal DIR values see our
comparison with the 2011 field campaign data set in Section 4.1.

The ice charts are saved as numerical grids from the ice charting software with a resolution of approximately 1 nautical mile (NM). In the grid format the ice thickness, ice concentration and DIR value assigned to each ice chart polygon are included. The size of a polygon is usually several hundreds of square kilometers. Additionally, the sea surface temperature is included in the ice chart grids. This practice slightly differs from the ice classes defined by the ice charting guidelines in Canadian Ice
Service, MANICE (2005), where the ice is classified based on the stage of development and indicated by the so-called WMO egg codes (WMO, 2010).

## 2.4 Surface and thickness profile data on ridged ice

No long-term studies between ice chart DIR and the actual ridging statistics have been published, although field campaigns to measure ridging in the Bay of Bothnia started in the late 1970's. First campaigns used shipborne laser profilers and the first extensive airborne laser profiler campaign was conducted in 1988 (Lewis et al., 1993). To compare DIR categories in the FIS charts with actual ridging we utilize data collected during helicopterborne profiling campaigns in the Bay of Bothnia. The main dataset is from the March 2011 campaign with approximately 600 km of measurement lines by a helicopterborne electromagnetic (HEM) sensor which combines laser surface profiling and inductive distance measurement to the ice-water interface. The measurement system was similar to that described by Haas et al. (2009). The HEM measurements give as comprehensive understanding on ridging as is obtainable from linear profiles (see Fig. 2). The two profiles provide the total thickness, and the surface laser profile resolves ridge sails. The measurement spacing of the HEM instrument is 3–4 m while the measurement response is obtained from a footprint which is typically 50 m. Standard inversion of the EM response assumes that the ice has uniform thickness and zero conductivity under the footprint. Neither holds for the roughly triangular ridge keels as their porous lower parts are permeable to electric currents. This results into underestimation of keel depths with 50 % or even more (Pfaffhuber et al., 2012) and also to underestimation of the total volume of ice.

The ice season 2010–2011 was severe, with a maximum ice extent of 309 000 km$^2$. In mid-basin of the Bay of Bothnia, the level ice thickness reached 60 cm with somewhat decreased ridging compared to the average winter with similar wind conditions.

To provide interannual variation for the HEM campaign based results of Section 4.1, we use data from the 1988 campaign and from three other campaigns in 1993, 1994 and 1997, summarized in Lensu (2003). The 1993 campaign was made in February and the others in March. They measured in total 1600 km of surface profiles. The 1988 campaign covers the whole Bay of Bothnia, the 1994 campaign covers it in the S-N direction while the 1993 and 1997 campaigns cover the NE quadrant of the basin. From the profile data ridge sails are selected with the Rayleigh criterion: to include the shallower one of two adjacent sails its height must be at least twice the minimum elevation between the sails. In addition, cutoff height is imposed. From the sail data the variation in ridge density, or the number of ridge sails per kilometer, and sail height can be determined. The ridge densities were 6.4, 7.3, 5.3 and 26.7 per kilometer for the 88, 93, 94 and 97 campaigns respectively. The the sail height shows less variation, from 0.58 m to 0.66 m. The densities are affected mostly by the number of days with strong winds during the earlier stages of the season when ice is less resistant to the deformation. The threshold wind speed for the onset of deformation is usually 14–16 m/s. The average values obscure the large regional variance in ridge density. It typically increases when moving towards north and east. Coastal ridge fields are often created by the closing of refrozen coastal leads and can be continuous rubble fields with densities up to 100 per kilometer. Sail height depends on the average ice thickness of the basin. Also the presence of snow reduces the heights in the profile data with a value equaling to the snow thickness. However, as a first approximation, the average height of sails exceeding 0.4 m can be assumed to be 0.6 m in the Bay of Bothnia. In the interpretation of the profile data it must be taken into account that a considerable fraction of ridges fall below the cutoff. Moreover, the sail heights are sampled from random crossings and include also shallower sections of the sails. In situ field

measurements usually select the highest point of the sail and the observed heights are typically 1–3 m and drilled keel depths 5–15 m (Kankaanpää, 1997).

## 2.5 Correspondence between ice charts and SAR mosaics

For a correct classification of the SAR texture features of sea ice, they need to be consistent with the degree of ridging values assigned for each class in the ice chart for the whole training data set. This consistency however cannot be assured by the current ice charting process, because of two main reasons. Firstly, the SAR data scenes are not always acquired over the area of interest in time for the ice charting (Rinne and Similä, 2016). This results in the ice analysts requirement for extracting ice information from other available sources, typically consisting in optical sensors (e.g. MODIS), in-situ measurements or observations (e.g. icebreakers). Secondly, the degree of ice ridging is divided into four severity classes. These ridging categories do not always have a clear separation between each other, thus in many cases they can be mixed.

To minimize the subjective bias and maximize the consistency between the texture features present in the daily SAR mosaics and the allocated ridging class in the corresponding ice charts, we used only those SAR - chart pairs which agree with each other on a daily basis to a decent degree and rejected all others. This was performed by examining every DIR chart during the whole test period from 1st January 2013 to 31st March 2013 and compare it visually to the corresponding SAR mosaic. To visually compare the sea ice texture features in SAR imagery and evaluate their correspondence to the correct ice class in the FIS charts is not a straightforward task when performed by a non-trained analyst. However, after inspecting several pairs visually and discussing with the ice analysts, the comparison was much easier and there were several cases when at least one major disagreement was found. Some of these disagreements arose and were confirmed to have appeared from the lack of fresh SAR imagery at the time of ice charting. In some other cases, the disagreements were clearly visible. If the DIR values assigned in the ice chart for a specific SAR texture region were consistent on muliple occasions, the exception pair was eliminated being considered inadequate for the classification. An example of the data selection procedure is presented in Fig. 3, where we show two SAR - chart pairs from 9th and 12th March 2013 over the Bay of Bothnia and northern part of the Bothnian Sea. Here, the two daily SAR mosaics show visually similar texture features with very small differences. In spite of that, the corresponding DIR charts show a change in class from slightly ridged ice of class 2 on 9th March to level-ice of class 1 on 12th March, along the west coast near the fast ice region. For ship navigation, the low ice ridging classes (DIR 1 and 2) do not likely pose any real concern. Therefore, if they are assigned differently in different days, the shipping is not affected much. On the other hand, for the automatic classifier this is a confusing case, that leads to a decrease in discrimination power between the two ridging categories. A similar effect can be seen in the more central to southern part of Bay of Bothnia, where the heavily ridged ice of class 4 on 9th March has changed to class 2 on 12th March. In this case, we have accepted only the data pair from 12th March as good pair, because this was found more consistent with the classification from multiple days (i.e. 13th, 14th March). In this case, the navigation in those areas might suffer.

In the end, our selection of daily SAR mosaics and FIS DIR charts pairs resulted in a total of 11 pairs in January, 15 in February and only 8 in March. All other data was rejected from the classification due to inconsistencies in at least one DIR class assigned to the corresponding SAR mosaic region or the time gap restriction (see Section 2.2).

## 3 Methodology for estimation of the degree of ice ridging

Our classification procedure consists of two stages. First, we segment the SAR imagery. The primary goal in the segmentation is that the resulting segments would mainly be composed of one DIR category. Then for each segment we compute a set of SAR texture features which are related to the ice ridging information. Their definitions are given below in Section 3.1. The
vector with the computed features as its components is called a feature vector. The second stage is to classify every segment based on the feature vector and assign one DIR category label to each segment. Hence, a successful classification requires that the segmentation succeeds, the features are meaningful and the feature based classification is efficient.

### 3.1 SAR Image segmentation

In order to perform the segmentation we combined the HH- and HV-polarized RS-2 SCWA images using the Principal Com-
ponent Analysis (PCA) technique. PCA is a statistical procedure that uses an orthogonal transformation to convert an image of possibly correlated pixel values into an image of linearly uncorrelated pixels. The values of these pixels are called principal component scores. We selected the first PCA image (corresponding to the largest PCA eigenvalue) for the segmentation because it explains most of the variation contained in the HH- and HV-images. This allowed us to speed up the segmentation process which for large SAR data sets requires a considerable amount of time and computing resources.

For the data segmentation, there are many algorithms available, but very few that can work well with SAR imagery because of the small dynamic $\sigma^o$ range. A clear separation between different sea ice types based on the magnitude of $\sigma^o$ does not exist, and the presence of the speckle complicates the segmentation task further. Here we have used a Markov Random Field approach (Rue and Held, 2005), and optimized with an Modified Metropolis Dynamics algorithm, similar as in Kato et al. (1992); Kato (1994) and Berthod et al. (1996). This stochastic method has been demonstrated to provide a better segmentation
than a deterministic one, e.g. Iterated Conditional Mode (ICM) by Besag (1974), for sea ice SAR imagery (Ochilov and Clausi, 2012; Deng and Clausi, 2004, 2005). The Markov Random Field (MRF) approach relies less on the initial segmentation than ICM, and also takes into account the global and local statistics of a pixel. This guarantees that pixels with similar intensities would not be treated in the same way in different regions of an image $I$, if the local spatial interactions differ in the two regions.

For example, to select the best new label $\hat{L}$ for a group of neighborhood pixels (clique) in a site $S$, is equivalent to maximize
the probability distribution of labels in the site, conditioned by the a-priori label ($L$) (Besag, 1974). In other words, $\hat{L}_{MAP} = argmax_{L \in \Omega} P(L|F = f)$, where $\Omega$ is the set of labels, $F = f$ is feature vector and $L$ is the segmented result conditioned by the feature vector. For each $S$, the cliques potential depends on the local configuration and type (size, shape, and possibly orientation). For simple cliques (formed by the closest neighbored pixels) in $S$, their potential function $V_c$ can be reduced to only two states:

$$V_c(L) = \beta \delta(L_i, L_j), \tag{1}$$

where

$$V_c(L) = \begin{cases} +1 & if \ L_i = L_j \ ; \\ -1 & if \ L_i \neq L_j. \end{cases} \tag{2}$$

and the homogenity of the region is controlled by the $\beta$ parameter.

The site's energy would simply be the sum of all cliques potential:

$$U(L) = \sum_{c \in C} V_c(L). \tag{3}$$

For more complex cliques (higher order neighbours), their potential would depend on the computed local mean ($\mu_{L_s}$) and variance ($\sigma^2_{L_s}$). The labels (classes) would then be represented by Gaussian distributions:

$$P(f_s|L_s) = \frac{1}{\sqrt{2\pi}\sigma_{L_s}} \exp\left(\frac{-(f_s - \mu_{L_s})^2}{2\sigma^2_{L_s}}\right) \tag{4}$$

If we consider the probability distribution of labels in $S$ a Markov Random Field with $P(L|F = f) > 0$, we can also treat it as a Gibbs form (Besag, 1974) :

$$P(L) = \frac{1}{Z} exp(-\sum_{c \in C} V_c(L)), \tag{5}$$

where $Z$ is the normalization constant and $V_c(L)$ is the clique's potential for the current label state.

For this example the (logarithmic) energy is

$$U(L) = \sum_S (\log((2\pi)^{1/2}\sigma_{L_s}) + \frac{(f_s - \mu_{L_s})^2}{2\sigma^2_{L_s}} + \sum_{s,r} \beta\delta(w_s, w_r). \tag{6}$$

Then the best new label for each site $S$ is determined by the minimization of its computed energy:

$$P(L|F = f) = \frac{1}{Z} \exp(-U(L)) \rightarrow \hat{L}_{MAP} = \text{argmax}_L P(L|F = f) = \text{argmin}_L U(L). \tag{7}$$

These kind of functions can be optimized by various methods, one being the simulated annealing method (Kirkpatrick et al., 1983) (Cerny, 1985), where a slow decrease in the probability of accepting worse solutions occurs as the algorithm searches the solution space. The method used here is an adaptation of the Metropolis-Hastings algorithm introduced in Metropolis et al. (1953) or shortly Metropolis algorithm which was created as a Monte Carlo method to generate sample states of a thermodynamic system. In the algorithm the labeling is also dependent on the control variable called temperature $T$. If the energy function U(L) value increases, the label is changed with a probability dependent on $T$ and increase of U(L) ($\exp(-\Delta U/T)$).

To perform the MRF segmentation, we first need to initialize the Gaussian parameters for the labels and also the number of labels. This is performed automatically for each SAR image separately. First the histogram of the SAR image is computed, and then the Expectation-Maximization (EM) (Dempster et al., 1977), algorithm is applied to decompose the histogram into a Gaussian decomposition. The number of Gaussians in the decomposition is initialized to a small minimum value, e.g. two, and

then iteratively increased until the EM decomposition and the histogram are similar enough with each other. We measure the similarity by the coefficient of correlation $r$. We stop the EM-algoritm if $r$ exceeds 0.97. The output number of classes possible for each image should also be limited. In our case we limit it to nine maximum allowed classes. We initially label the image pixels based on the EM classification, i.e. we assign the label with highest probability of the N different Gaussian distributions $G_k(x)$ for a pixel. After this labeling scheme we can run the MRF segmentation.

Here, we have applied the segmentation on the first PCA component image. An example of segmentation result for the Bay of Bothnia region is shown in Fig. 4 and Fig. 11 together with the original HH and HV SAR mosaics.

The next step in the SAR analysis is to compute several SAR statistics (features) for the obtained segments.

## 3.2 SAR image features

We studied the classification of DIR categories using the computed SAR features and the DIR values from FMI ice charts. The following SAR features were computed from the SAR images with 100 m pixel size, and their efficiency in the DIR classification was studied. Each feature value is a median value of the feature computed over a single segment.

1. HH polarization SAR backscattering coefficient ($\sigma_{HH}^o$), with incidence angle correction applied.

2. HV polarization SAR backscattering coefficient ($\sigma_{HV}^o$), with incidence angle correction and noise level equalization applied.

3. HH entropy ($E_{HH}$), computed in windows with a radius of 5 pixels.

4. HV entropy ($E_{HV}$), computed in windows with a radius of 5 pixels.

5. HH autocorrelation ($AC_{HH}$), computed in windows with a radius of 5 pixels.

6. HV autocorrelation ($AC_{HV}$), computed in windows with a radius of 5 pixels.

7. HH coefficient of variation ($CV_{HH}$), computed in windows with a radius of 5 pixels).

8. HV coefficient of variation ($CV_{HV}$), computed in windows with a radius of 5 pixels.

9. Edge density for HH image($ED_{HH}$), scaling: $1000 * N_e/A$ ($N_e$ is the number of edge pixels and A is the segment area).

10. Edge density for HV image($ED_{HV}$), scaling: $1000 * N_e/A$ ($N_e$ is the number of edge pixels and A is the segment area).

11. Segment size (SSZ).

12. HH kurtosis ($K_{HH}$), computed in windows with a radius of 5 pixels.

13. HV kurtosis ($K_{HV}$), computed in windows with a radius of 5 pixels.

Additionally we extracted the segment mean of sea ice concentration (SIC) from the FMI ice charts.

Coefficient of variation was computed separately as

$$C^V = \frac{\sigma}{\mu}, \tag{8}$$

where $\sigma$ is the standard deviation and $\mu$ is the mean over the window. Kurtosis is computed as the fourth moment within the data window.

Entropy $E$ (Shannon, 1948) was computed as

$$E = -\sum_{k=0}^{255} p_k log^2 p_k, \tag{9}$$

where $p_k$'s are the proportions of each gray tone $k$ within a window. Auto-correlation, $AC$ (Similä, 1994; Karvonen et al., 2005), was computed as

$$C^A(k,l) = \frac{\sum_{ij \in B} \left(I(i-k, j-l) - \mu_B\right)\left(I(i,j) - \mu_B\right)}{|B|\sigma_B^2}, \tag{10}$$

where $I(k,l)$ is the pixel value at location (k,l). Mean over the directions horizontal, vertical and diagonal directions i.e. $(k,l) = (0,1)$, $(k,l) = (1,0)$, $(k,l) = (1,1)$ and $(k,l) = (1,-1)$ was used to accomplish directional isotropy. The computation window is denoted by B.

Edge density $D$ was computed for each segment (separately at the HH and HV polarizations) after an edge detection by the Canny algorithm (Canny, 1986) as

$$D = N_e/N, \tag{11}$$

where $N_e$ is the number of edge pixels with a segment and N is the segment size in pixels.

Most of the features have a straightforward interpretation. Entropy describes how uniformly the HH/HV values are distributed. Edge density is a measure for edge fragments present in the segment which we assume to be related to ridging. Coefficient of variation (CV) describes how fast the standard deviation increases with the mean. We expect that in the ridged areas CV is larger than in the homogeneous areas. Kurtosis describes the peakiness of the $\sigma^o$ distribution. With the aid of the spatial autocorrelation we can quantify how structured the ice field in question is in the SAR imagery. We expect that more structural elements appear in the ridged ice than in the level ice where the spatial $\sigma^o$ variation is more random.

In Fig. 5 the computed features for an area in the central Bay of Bothnia on 15[th] March 2013 are shown. Moderately and heavily ridged areas were present here.

## 3.3 Random forest classification method

After trying several classification methods (local regression, logistic regression, General Additive Regression Model) we found that the random forest (RF) (Breiman, 2001) approach produced good enough results to be of practical use. Random forest is an ensemble learning method which can be applied to classification and regression. In RF we artificially generate several training

sets from a single training set at our disposal using bootstrapping, grow a classification tree for each individual training set, perform classification for each tree and then aggregate the results. The bootstrap aggregation is called bagging. This technique is efficient to reduce variance in high-variance predictions in the same manner as taking an average of samples in Breiman (2001).

For the classification of the daily sea ice data we divided our data into the training and the test data sets (see Section 2.2). In our computations we have used the routines included in the commercial software Matlab.

### 3.3.1   Description of the algorithm

We outline here the RF classification method and the notations used in this algorithm. The classes are denoted by $C = \{1,\ldots,C\}$ We have a training set $X = \{x_1,\ldots,x_N\}$ where each sample $x_i$ consists of a feature vector $f_i$ and the corre-

sponding class. When we take a bootstrap sample from $X$, we denote it $Z^*$. Our bootstrap sample $Z^*$ is of the same size as the original sample, so on average the fraction 63% of the original samples of $X$ belong to it, the rest being duplicates (Efron and Tibshirani, 1993). The samples of $X$ left out from $Z^*$ (about 37% of the samples) are called out-of-bag (OOB) samples.

The classification tree is denoted $T_b(\Theta_b)$ with $b \in \{1,\ldots,B\}$ and it uses $Z^*$ as its training data. Each end node $n$ of $T_b(\Theta_b)$ has a class label which is the most frequent class in that node. The parameter $\Theta_b$ characterizes the $b$th random forest tree in

terms of split variables, split points at each node, and terminal node class label. The class label given by $T_b(\Theta_b)$ depends on the feature vector $f_i$ which is used as input for the tree. It is denoted as $\hat{C}_b(f_i,\Theta_b)$. We generate $B$ bootstrapped training sets and relying on every training set we grow a classification tree $T_b(\Theta_b)$. A classification tree often achieves a rather low bias if it is grown deep with many nodes without pruning (Hastie et al., 2011) .

The impurity measure is the Gini index $G$,

$$G = 1 - \sum_{c=1}^{C} p(c|n)^2$$

where $p(c|n)$ is the proportion of the samples that belong to class $c$ at a particular node $n$. $G$ indicates how dominant the class $c$ is in the subtree after the split. A small Gini index value indicates that the subtree contains predominantly observations from a single class. In the split the feature component of the vector $f_i$ with the smallest Gini index is utilised (Ripley, 1996).

In classification we record the classes predicted by the ensemble of $B$ trees for a specific feature vector, and take a majority

vote. The most common class is the class predicted by the ensemble. Then the selected class has a smaller uncertainty than a single classification tree (Hastie et al., 2011), because an average has a smaller variance than a single variable. This is true also for the correlated variables. If $B$ is large enough then the random forest algorithm avoids the tendency of over fitting the model which often occurs in the context of the decision trees.

The problem with bagging is that the grown trees are correlated. To reduce this correlation the RF has a randomisation step.

When building trees, each time a split in a tree is considered, a random sample of $m$ predictors is chosen as split candidates from the full set of $p$ predictors ($m = 4$ and $p = 8$ here). A new sample of $m$ predictors is taken at each split. This step prevents the same features from dominating every tree.

The flow of the random forest algorithm is described below.

**Random forest algorithm for classification**

1. For $b = 1$ to $B$:

    (a) Draw a bootstrap sample $Z^*$ of size $N$ from the training data.

    (b) Grow a random-forest tree $T_b(\Theta_b)$ with the bootstrapped data, by recursively repeating the following steps for each terminal node of the tree, until the minimum node size is reached.

    i. Select $m$ variables at random from the $p$ variables.

    ii. Determine the best variable and split-point among the $m$ variables using the Gini index.

    iii. Split the node into two daughter nodes.

2. Output the ensemble of trees $\{T_b(\Theta_b)_1^B\}$.

To classify a new feature vector $f_n$:

*Classification*: Let $\hat{C}_b(f_n, \Theta_b)$ be the class prediction of the $b$th random-forest tree. Then $\hat{C}_{rf}^B(f_n) = $ *majority vote* $\{\hat{C}_b(f_n, \Theta_b)\}_1^B$.

### 3.3.2 Selection of the features

Because an ensemble of trees was used in RF and a large amount of features were utilized, the results were hard to interpret. To analyse the impact of different features on the class estimation the importance measure was used. The value of the importance measure is called an importance value. This measure is implemented as follows:

For each tree, the classification error on the OOB portion of the data is computed. This gives the baseline error rate for the tree. Then in the OOB set we randomly permute one feature of the feature vector $f_i$ and simultaneously keep fixed the other features in $f_i$. We note that the marginal sampling distribution of the picked feature remains the same during the permutation. Next, we recalculate the classification error in the OOB set. This classification error is compared to the baseline error. Usually it is larger than the baseline error. The procedure is repeated for every feature separately. The decrease in classification rate as a result of this permuting is averaged over all trees, and is used as a measure of the importance of the chosen feature.

To select the features we run the RF algorithm for several feature combinations and for several different training data sets. The importance of the features as well as the classification accuracy was monitored. This empirical approach lead to the choice of 8 features from the computed 13 features introduced in Section 3.2 including the additional SIC feature.

In summary we found that the RF classification presents the following advantages : i) RF has the ability to describe complex, nonlinear statistical relationships among variables, ii) RF reduces the uncertainty of the obtained estimate and iii) RF reduces

the possibility of over fitting. The greatest weakness in RF is its relatively weak extrapolation property (Hastie et al., 2011). This property can be seen from the behaviour of the error rates. The RF classifier has a very low training error rate but the error rates increase significantly for the test set.

## 4   Results

### 4.1   Ice chart ridging categories vs. surface and thickness profile data

We use the March 2011 HEM campaign data to study how well the DIR categories in the FIS charts describe the actual ridging. As the variation is much larger for ridge density than for sail height, ridging is parameterized here by density only. The other parameter considered is the total thickness of ridged ice. This is a navigationally relevant parameter that can be used to calculate icegoing speed of ships. To establish compatibility with the ice chart, which employs a 1x1 nautical mile (NM) grid, the sail density and ridged ice thickness from the March 2011 data were calculated as averages for the cells of the grid. There are two different aspects of comparison involved. The first is the relationship between ridge density and DIR in the ice chart data. The other is the relationship between ridge density and ridged ice thickness which is in general relevant to the question how well surface data can represent the total thickness of ice. Thus, the comparison is made between ridge density and DIR and, on the other hand, between ridged ice thickness and DIR. Although somewhat qualitative the DIR indices are estimates made by sea ice specialists and refer to the Lagrangian ice chart regions corresponding to various formation and deformation phases of the ice cover. The reliability or their boundaries is usually high. The DIR value 1, corresponding level and rafted ice categories, had very small coverage in the data, while DIR 2, the category of slightly ridged ice, was not found at all. The comparison is therefore made for the DIR values 3 and 4, or moderately and heavily ridged ice. The sail height retrieved from the profile data was equal for these categories while a clear difference was found for the ridge densities and ridged ice thicknesses, see Table 1. This indicates that a rough but reliable quantification of ridging can be based on DIR values only. A more detailed picture can be obtained from comparisons of Fig. 6 between DIR and, on the other hand, ridge density and total ice thickness from the March 2011 data. For the ridge density the colorbar range is chosen to be from 12.7 to 21.5 which are the average densities corresponding to DIR 3 and 4 in Fig. 6. Ice thickness colorbar was scaled similarly. Thus, all values below the averages corresponding to DIR 3 are blue and all values above averages corresponding to DIR 4 are pink. Above and below the colorbar range the ridge density has still a wide range of variation as is seen from the histogram in Figure 6. However, the basic regional characteristics are similarly visible in all three datasets. In spite of the uttermost simplicity of DIR it was in a reasonably good agreement with both ridge density and total thickness. The agreement with DIR was somewhat better for the total ice thickness than for the ridge density. This may be related to the fact that a large fraction of ridging does not show in the density due to the cutoff but affects the SAR-based and visual estimates behind DIR values. The generally good agreement between DIR, ridge density and total thickness means that DIR values estimated from SAR can be translated to navigationally relevant ridging or thickness parameters. The largest differences between the degree of ice ridging and HEM quantities were found in the coastal ridge field extending from 64° N 23° E towards SW (see Fig. 2). Both ridge density and average sail height were lower for this part in comparison with the extension of the same ridge field towards NE from the said

location. These values were also similar to those found in the mid basin, so the missing separation of this coastal ridge field into two categories apparent from the HEM data is clearly a shortcoming of the ice chart DIR data.

## 4.2 Monthly backscattering statistics

We concentrated our analysis on the areas with SIC over 80 %. In areas with ice concentration varying from 80 % to 90 %, the amount of open water can impact the backscattering statistics significantly, particularly during high winds. This SIC limitation excluded the marginal ice zone (MIZ) which is defined to consist of ice areas with SIC from 15 % be 80 %, see e.g. Strong (2012), from our analysis. Almost all areas with SIC from 80 % to 90 % belonged to level ice polygons in our data set. In total the level ice category DIR 1 covered well over 50 % of all the ice areas during our test period.

According to earlier studies the effect of incidence angle on $\sigma^o$ for level ice and ridged ice is rather similar. In Mäkynen et al. (2002) it was found that the incidence angle dependence of $\sigma^o_{HH}$ in logarithmic scale (dB) can be described by a linear model, with slopes -0.21 dB/degree for ridged ice and -0.25 dB/degree for level ice. It seems that using a slope of -0.23 for all the data is adequate for automated classification, and the ridged areas and level ice can be distinguished both at near and far range. Also a more sophisticated approach, iteratively applying different slopes for level ice and ridged ice has been studied in Karvonen et al. (2002), but the effect on sea ice classification was minor. When inspecting the SAR mosaics visually most of the SAR frame boundaries were not visible or were hardly visible, indicating successful $\sigma^o_{HH}$ incidence angle correction. For open water the correction may not work properly as for open water $\sigma^o$ signatures depend heavily on wind speed and swell (i.e., surface roughness).

For HV channel, the combined incidence angle and noise floor correction is essential. Without this correction the HV backscattering and texture features derived from it can not be used in classification as the effect of the varying noise floor is so high (up to about 3 dB) and will cause a significant amount of misclassifications. However, after correction the HV channel data can be used in classification and we have not by visual inspection observed any significant differences in near-range and far-range $\sigma^o_{HV}$ for either open water, level ice or ridged ice classes.

We first looked at how the $\sigma^o_{HH}$ distribution changes monthly from January to March 2013 in two main ice categories: level ice (DIR 1) and ridged ice (DIR>1) (see Fig 7). In the beginning of January level ice appeared mainly near the coast of the Sea of Bothnia, and the dominance of mostly thin ice in the Sea of Bothnia continued up to the middle of February whereas in the Bay of Bothnia appeared ridged ice areas throughout the whole test period. A significant fraction of the level ice pixels had $\sigma^o_{HH}$ value below -18 dB indicating thin smooth ice. For thin ice relatively high $\sigma^o_{HH}$ (over -18 dB) have also been observed (Mäkynen and Hallikainen, 2004). Also, the presence of the level ice areas with a relatively low SIC (80–90 %) meant that open water patches affected the level ice $\sigma^o_{HH}$ yielding both high and low $\sigma^o_{HH}$ values. The level ice $\sigma^o_{HH}$ values above -18 dB were evenly distributed in January. Ridged ice areas had a large $\sigma^o_{HH}$ peak at -16.5 dB and most of the remaining pixel values ranged from -16 dB to -12 dB.

In February level ice $\sigma^o_{HH}$ still had a sharp peak around -20 dB indicating presence of very thin ice, but most of $\sigma^o_{HH}$ values had spread between -16 dB and -11 dB. In the ridged areas a majority of $\sigma^o_{HH}$ values were in the range between -17 dB and -12 dB. The mean and median values of $\sigma^o_{HH}$ for level ice areas and ridged areas were almost identical in January and

February. In March the $\sigma_{HH}^o$ statistics for level and ridged areas showed here a clear discrepancy. The level ice $\sigma_{HH}^o$ values were distributed from -20 dB to -10 dB whereas the $\sigma_{HH}^o$ values from ridged areas were concentrated in the range from -15 dB to -11 dB. The average $\sigma_{HH}^o$ value from the ridged areas was over 2 dB higher than that of the level ice areas unlike in previous months. There was a significant increase in the magnitude of $\sigma_{HH}^o$ from ridged areas whereas this was not the case for the level

ice $\sigma_{HH}^o$.

Based on Fig. 7 it was obvious that the magnitude of $\sigma_{HH}^o$ alone could not be an efficient predictor in the estimation of the DIR value in January and February.

We assume that the small separation in the $\sigma_{HH}^o$ values originating from level ice and ridged ice during the first two test months was due to following two main reasons. As mentioned earlier the level ice areas had mostly SIC less than 90 %. So the

backscattering from open water had a significant contribution to the level ice $\sigma_{HH}^o$. In addition, the level ice area (DIR 1) had significant uncertainties in the FMI charts. The ice analysts responsible for the charts told us that in several cases it was difficult, nearly impossible, to discriminate reliably between level ice and slightly ridged ice (DIR 2). In these cases they usually chose the level ice category if the icebreaker reports did not indicate any difficulties for merchant ships. If these had been reported, a slightly ridged ice category (DIR 2) was chosen.

Considering the $\sigma_{HH}^o$ contrast between level and ridged ice areas, the situation changed gradually in February and March towards a more distinct separation between these ice types. We shall analyse the DIR charts separately for the period of strong thermodynamic ice growth (January–February) and more stable winter conditions (March) in Section 4.3 .

The examination of the monthly $\sigma_{HV}^o$ distributions, see Fig. 8, confirms our findings for the $\sigma_{HH}^o$ distribution. In January and February the values of $\sigma_{HV}^o$ from level ice areas were close to the noise floor (-28 dB) and hence uninformative for a

meaningful analysis. In the ridged areas $\sigma_{HV}^o$ were 2–3 dB higher but still in general rather low indicating a low sea ice surface roughness. In March the $\sigma_{HV}^o$ distributions both in level ice areas and ridged areas were on average about 1–2 dB higher than in the previous months. Also the contrast in the $\sigma_{HV}^o$ between ridged and level ice was more significant. In consequence, the $\sigma_{HV}^o$ values affected the classification result in March but were not useful in the earlier months.

### 4.3   Classification results for several ridging categories

There was a fundamental imbalance between the sample sizes representing level ice and ridged ice classes. The samples from all the ridged ice classes formed about 40 % of all samples. If we had required that all the classes were of equal size in the training data, the amount of observations per ice category would have been low, e.g. less than 20 % of the level ice samples would have been utilized. When assessing the results we will keep in our mind the highly different sizes of the ice classes.

We run all our random forest classifications with the same set of tuning parameters for routine TreeBagger (Matlab, 2016).

From the set of eight ($p = 8$) features we randomly chose $m = 4$ features to be used in a split. Often the value $m = \sqrt{p}$, i.e. $m = 3$ here, would have been recommended (Hastie et al., 2011). However, we noted that slightly better results were obtained with $m = 4$ for our data set. Another fixed option was that the minimum number of data points in the end nodes was set to ten. We grew 200 trees during the classification. Results with more decision trees did not yield any significant improvement of the error rates.

In the first phase we investigated the capacity of the RF classifier to separate between level and ridged ice. Data of all three months were included in the analysed data set. The results are presented in Table 2. The overall classification rate was 82 % for the whole winter.

Next we examined the classification of all the four ridging categories through the three-month period. The training and test data sets had been selected from each month in our data set. The overall classification rate for the test period was 71 %. Looking at the confusion matrix in Table 3 we can observe that the level ice category (93 %) had a very high classification rate. The classification of the three categories for ridged ice was more challenging. The ridged ice category (DIR 3) was classified correctly in 45 % of the cases but over 30 % of the observations were confused with level ice. The slightly ridged ice (DIR 2) was poorly distinguished. Only in the 15 % of the cases is was detected correctly. Most DIR 2 samples (42 %) were assigned to the level ice category which in the light of the previous discussion could be expected, i.e. the preference among the FIS ice analysts to use level ice category over slightly ridged ice in the manual ice charts. The ridged ice category with the most accurate classification rate (59 %) was the heavily ridged ice category (DIR 4).

To obtain more information on how the adopted approach works in rapidly changing ice conditions and on the other hand in more stable winter conditions we classified all three test months separately so that the training and testing data were collected during the same month. The overall accuracy of the monthly results varied largely being at its lowest in February (63 %), higher in January (83 %) and March (81 %). The corresponding Cohen's kappa figures were 0.60 (substantial agreement), 0.52 (moderate agreement) and 0.68 (substantial agreement). The separation between all ice categories was the best in January (overall accuracy 83 %) where basically only three DIR categories appeared. An evidence that the definition of different DIR categories were inconsistent with each other in January and February was that in these months the detection rate stayed below 100 % for the training data in the RF classification but it was 100 % for the March training data.

In each month level ice was the dominant ice category being over 50 % of all ice covered area. The DIR 2 category covered from 6 % to 19 % of the ice area depending on the month. In none of the months was DIR 2 successfully detected due to its ambivalent definition with respect to the DIR 1 category. The DIR 3 category was successfully detected in January when its areal coverage was large (21 %), and in March when the boundaries between different ice categories were best defined during the test period. The heavily ridged ice fields (DIR 4) were classified well, except in January when such ridged areas were rare (about 1 %). A possible explanation for the lowest accuracy rate in February was that then the boundaries between different DIR regions were often visually rather difficult to discern in the SAR imagery according to our experience. Figure 9 shows the variation of the detection rate for each DIR category in all the classification results. The most distinct feature in the results is the consistently poor detection rate for DIR 2.

In Fig. 10 we can see the Baltic Sea ice DIR classification result (left) for a dual-polarized SAR image mosaic on 9[th] February 2013 (Fig. 4 top). Also the reference DIR chart is shown for comparison (right). The automated DIR chart produced agreed well with the FIS ice charts for DIR values 1 and 3. However, the automated chart estimated a large fraction of DIR 2 category ice to DIR 4 category. The automated DIR chart contained detailed markings of the cracks and openings in the central Bay of Bothnia which were not present in the FIS chart. We remark that the SAR mosaic on 9[th] February looked very similar to the one on 7[th] February (two days earlier), when the same cracks / openings can be found, but the corresponding FIS ice chart DIR

showed DIR 4 in the areas to which was now assigned DIR 3. This can be taken as an example of the subjectivity which is inherent to the manual ice charts.

There is a good overall agreement between the FIS chart and our DIR classification in Fig. 12. Most of the differences occur in the Bothnian Sea. There the FIS chart indicates mostly level ice and to some extent slightly ridged ice. On the other hand, the classification assigned to some FIS level ice areas the ridged ice and heavily ridged ice categories. Based on the SAR HH- and HV-polarization mosaics (see Fig. 11) those areas represent broken ice fields although the ridging intensity is hard to assess visually.

The major reason for the success of the classification in March is the better discrimination between the ridged ice and level ice in March than in the previous months as noted earlier in Section 4.2. The better discrimination property between ridging ice categories affects the final results in two ways. First, the segment boundaries of the dual-pol SAR imagery follow better the boundaries of the DIR classes in March (see Fig. 11). Secondly, the segmentwise feature vectors show more variability between different ridging categories in midwinter. The combination of these two factors determine the accuracy of the final classification.

We studied the success of the segmentation by examining how large fraction of the segments contained practically just one ridging category. i.e. the area of some ridging category covered over 90 % of the segment area. The results were that in January 93 % of the SAR imagery belonged to such segments, in February 80 % and in March 86 %. The high fraction of well defined segments in January is easy to understand because most of the ice was level ice (72 % of the area), and just three ridging categories appeared (the heavily ridged area covered less than 1 %). In February the fraction of level ice has decreased to 55% of the total area, all four ice categories were present and the total area of well defined segments decreased to 80 %. In March the level ice area covered 59 % of the total area and the area of the well defined segments was 86 %. Hence there was better segmentation accuracy in March than in February. In that month the total area of correctly classified ridging categories was 81 %, five percent points less than the total area of the well defined segments. In February the total area of correctly classified ridging categories was just 63 % which means 17 percent points less than the total area of the well defined segments. This analysis suggests that the main separating factor contributing to the classification accuracy was due to the more versatile feature vectors in March.

### 4.4   Importance of features

The selection of the eight features in Section 3.3.2 was based on their importance value. The features consisted of six HH-polarization based segment-wise features (see Section 3.2) and the segment-wise $\sigma_{HV}^{o}$ as well as the SIC value extracted from the FIS ice chart. Their importance order when the training data covered the whole test period is presented in the Table 4. If the training data of just one month was used the importance order of features varied slightly. The importance of one specific feature is relative in the sense that it changes when the combination of the used features changes, i.e. the importance of one feature depends on which other features are included. The feature SIC remained however the most influential feature in every case. This is comprehensible because when SIC was between 80 % and 90 %, the ice area in question represented almost always the

level ice category (DIR 1) and the corresponding feature vector was easy to classify correctly. The rather low importance value of $\sigma_{HV}^o$ is probably due to the relative narrow range of the $\sigma_{HV}^o$ values.

To gain more insight into how the eight selected features affected the classification accuracy, we studied the possibility of feature reduction using the March data as benchmark. The March data was selected because the diversity of ridging categories was largest then (see Table 3). We systematically eliminated the selected features one by one and reclassified the March test data using the remaining features. In none of the cases did the classification accuracy improve with fewer features. For several removed features ($E_{HH}$, $AC_{HH}$,$K_{HH}$, $\sigma_{HV}^o$ ) the classification accuracy decreased by just a few percent points (1–3 %). The removal of the $ED_{HH}$ feature did not practically affect the accuracy at all. A significant misclassification rate increase was observed by the reduction of the $\sigma_{HH}^o$ (-6%), $CV_{HH]}$ (-8%) and SIC (-12%). In every case the relative importance of the retained features changed. Hence the importance of the features present in Table 4 is true only in the context of this specific feature combination.

To see more clearly that the features included in the feature vector complement each other and make the classification more robust, we classified the March data using only three basic features ($f_3 = (SIC, \sigma_{HH}^o, \sigma_{HV}^o)$). The overall accuracy was just 64 %. Then we added the feature $CV_{HH}$ to $f_3$ because $CV_{HH}$ caused a significant drop in the accuracy. The accuracy remained low, only 68 %. Our conclusion is that the information provided by the whole feature set is needed for a good description of ridged ice field in the SAR imagery. If already a reduction of one feature decreases the classification accuracy, the reduction of two or more features would degrade the classification further. The only feature which may be unnecessary is $ED_{HH}$. It was also the most heuristic one (see Section 3.2). Because it does not decrease the classification accuracy, we kept it in our feature combination. We also experimented replacing the HH-polarization based features with their HV-polarization counterparts. This lead to degradation of the classification accuracy in all of the studied cases.

## 5   Discussion and Conclusions

The degree of ice ridging is one of the most useful parameters for ice navigating ships. It basically indicates, together with ship characteristics, whether a vessel can safely pass through an ice field or not. The DIR also complements the more general Risk Index Outcome (RIO), defined by IMO (2016), as this does not address ridging but relies on WMO categories for the stage of development. We have shown that an automated estimation of the DIR from SAR texture features, together with an ice concentration estimate, performs well when compared to the values extracted from the manual FIS ice charts. The applied features describe statistics of $\sigma^o$ variation in the SAR imagery. DIR estimation is a suitable task for a SAR based approach because the C-band $\sigma^o$ is sensitive to the large scale surface roughness due to ridging.

Both independently operating ships as well as ships relying on icebreaker support operate in ice infested waters. In the Baltic Sea most of the merchant ships need icebreaker assistance. However, ships of the highest Finnish-Swedish ice class in the Baltic Sea, 1A Super, which is equivalent to the Polar Class PC 6, are designed to operate in difficult ice conditions independently. The FIS ice charts are prepared to serve operations where ships follow an icebreaker in a convoy. Based on discussions with the FIS ice analysts the following remark is made. If the ice conditions in an area do not pose a realistic

risk for icebreakers to get stuck, a smaller DIR value is often assigned to this area even if the area is difficult for independent navigation of merchant ships. This is especially true for DIR 2. Hence, the availability of the icebreaker assistance has an effect on the DIR classifications in the FIS ice charts.

The primary objective of our DIR classification algorithm is to separate the severe ice conditions from the easier navigable ones. To reach this goal our DIR classification mainly relies on SAR image statistics. In some cases this may lead to differences between the FIS ice charts and our classification results because the FIS charts take the icebreaker factor into account, which is not present in the SAR imagery. Hence, these two data sets can be interpreted from slightly different perspectives. An example of this difference is our earlier discussion related to Fig. 10 and Fig. 12. One of the essential advantages of the automated DIR charts is that they include leads and small level ice areas between ridged areas not present in the coarser FIS charts.

We used a two-stage classification system. First, we segmented the dual-pol SAR mosaics. This succeeded slightly differently for different months. The area of level ice always exceeded 50 % of ice cover. In January it was highest, over 70 %. In that month 93 % of the segmented area belonged to the segments dominated by one ice category. In February and March the respective figures were 80 % and 86 %. It should be noted that for January only three DIR categories were present, unlike for the last two months where all four DIR categories appeared. We can conclude that the SAR signatures matched the DIR boundaries best in March when the amount of ridging in our test period was at its maximum.

In the second phase of the classification we classified the segments using segment-wise feature vectors, classifying each segment to one ridging category. This succeeded best in March (82 %). Then the ridging intensity varied largely in different regions in our test area and the resulting texture of the SAR imagery was more versatile than in the other studied months. It is worth noting that in March the accuracy of the feature based classification was just five percentage points lower than the total area of the well-defined segments, i.e. the feature based classification succeeded with the RF classifier. This result can be regarded as a confirmation that the computed features were well suited to describe the ridging in the SAR imagery. In January the classification accuracy was at the same level as in March (83 %) but the area covered by the well-defined segments was much larger, indicating that the feature based classification did not perform as well as in March. In January and February the ice was thinner and the degree of ridging lower than in March. In these two months the $\sigma_{HH}^{o}$ and $\sigma_{HV}^{o}$ distributions from level ice and ridged ice overlapped substantively. This weak discrimination between level ice and ridged ice can be partly attributed to the subjective interpretation of the level ice category at FIS as discussed in Section 2.5.

Our approach works best in the Baltic Sea when the evolution of winter has passed the freezing phase and a significant amount of ridging has occurred. Then ridging strongly contributes to the texture of the SAR images.

Before setting up an operational DIR estimation system over the Baltic Sea, we need to test our algorithm with more winters data and to optimise it for the best possible result. In an operational mode we can use the most recent SAR/FIS and SIC data for the training. Instead of using the SIC information present in the FIS charts we can also use an automated radiometer or combined radiometer/SAR based SIC data. Currently, the finest resolution in operational SIC products is offered by the Advanced Microwave Scanning Radiometer 2 (AMSR2) based ASI sea ice algorithm (Beitsch et al., 2014). The grid size in the product is 3.125 km. To improve our product during ice forming or melting periods, we can include ice thickness as an additional parameter in the future DIR classifications.

We have plans to extend our algorithm to the Arctic Ocean, where there is a high demand for reliable ice information for independently navigating merchant vessels. Harsh ice conditions, as those in March 2013 in this study, prevail much longer in the Arctic seasonal ice regime than in the Baltic. The consistent availability of DIR charts in Arctic would enable the monitoring of areal evolution of different ridging intensity categories. An automated DIR chart utilizing fine resolution SAR

data and classifying the suitability of different areas for navigation would benefit all Arctic shipping.

The ice ridging and its backscattering mechanisms are similar in the Baltic and Arctic. In general, Arctic sea ice is thicker and ridges can be larger than in the Baltic. However, the backscattering increases as a function of the surface roughness in both areas. We have visually inspected SAR imagery over seasonal sea ice in Barents and Kara seas and the look similar than corresponding imagery over the Baltic. This builds confidence that our algorithm, with possible minor adjustments, should be

applicable to the Arctic first year ice. Nevertheless, the possible applicability of the method to the multi-year ice areas must be studied separately.

*Data availability.* All data can be obtained by contacting the first author.

*Author contributions.* The concept of the study was conceived by AG, JK and MS. The SAR processing was done and reported jointly by JK and AG. MM commented the SAR processing. The statistical analysis was performed and written by MS. ML analyzed and wrote about the

15 2011 field data. MM and AG wrote the introduction. AG and MS wrote the conclusion section. JV commented the results and the conclusions from the point of view of an ice analyst. All authors contributed to the editing of the text.

*Competing interests.* The authors declare that they have no competing interest.

*Acknowledgements.* Firstly, the authors wish to thank the editor and the reviewers for the constructive comments and suggestions provided, which greatly improved the final version of the paper. We also thank Eero Rinne for proofreading the paper and making valuable com-

20 ments. The work of Alexandru Gegiuc, Juha Karvonen and Marko Mäkynen was supported by Space-borne observations for detecting and forecasting sea ice cover extremes (SPICES) funded by the European Union's Horizon 2020 Programme (Grant Agreement No. 640161). Markku Similä and Mikko Lensu were supported by the KAMON project funded by Academy of Finland (contract 283034) and the BONUS STORMWINDS project funded jointly by the EU and the Academy of Finland (contract 291683).

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

**Table 1.** Comparison of ice ridge statistical parameters derived from the HEM data with ice chart degrees of ice ridging.

| DIR | 3 | 4 |
|---|---|---|
| Number of cells | 590 | 1079 |
| Ridge sail height [m] | 0.61 | 0.61 |
| Ridge density [1/km] | 12.7 | 21.5 |
| Total thickness [m] | 0.76 | 1.08 |

**Table 2.** Confusion matrix for the level ice vs. ridged ice (categories DIR 2 to 4) in the RF classification for the whole test period.

| FIS | Sample size | Ice category | |
|---|---|---|---|
| Ice cateogory | N | level | ridged |
| level | 59 % | **87 %** | 13 % |
| ridged | 41 % | 24 % | **76 %** |

**Table 3.** Confusion matrix for all ridged ice categories in the RF classification.

| | FIS | Sample size | RF classes | | | |
|---|---|---|---|---|---|---|
| | DIR | N | 1 | 2 | 3 | 4 |
| whole period | 1 | 59 % | **93 %** | 1 % | 4 % | 2 % |
| | 2 | 14 % | **42 %** | 15 % | 28 % | 15 % |
| | 3 | 15 % | 32 % | 5 % | **45 %** | 18 % |
| | 4 | 12 % | 18 % | 3 % | 20 % | **59 %** |
| January 2013 | 1 | 72 % | **94 %** | 2 % | 4 % | 0 % |
| | 2 | 6 % | **48 %** | 15 % | 36 % | 0 % |
| | 3 | 21 % | 35 % | 1 % | **64 %** | 0 % |
| | 4 | 1 % | **47 %** | 0 % | 21 % | 32% |
| February 2013 | 1 | 55 % | **92 %** | 1 % | 5 % | 2 % |
| | 2 | 19 % | **46 %** | 13 % | 21 % | 20 % |
| | 3 | 17 % | 21 % | 6 % | 29 % | **44 %** |
| | 4 | 9 % | 18 % | 16 % | 16 % | **51 %** |
| March 2013 | 1 | 59 % | **92 %** | 4 % | 2 % | 2 % |
| | 2 | 9 % | 16 % | **32 %** | 21 % | 31 % |
| | 3 | 10 % | 44 % | 4 % | **51 %** | 1 % |
| | 4 | 22 % | 4 % | 2 % | 6 % | **88 %** |

**Table 4.** The importance of different features when the training data covered the whole test period.

| feature | SIC | $K_{HH}$ | $\sigma^o_{HH}$ | $ED_{HH}$ | $AC_{HH}$ | $E_{HH}$ | $CV_{HH}$ | $\sigma^o_{HV}$ |
|---------|-----|----------|-----------------|-----------|-----------|----------|-----------|-----------------|
| importance (%) | 13.9 | 11.9 | 11.7 | 11.3 | 8.1 | 7.2 | 7.2 | 6.9 |

**Figure Captions**

Figure 1. Example of RS-2 dual polarized SAR image mosaic (left: HH, middle HV) over the Baltic Sea on 15[th] March 2013 and the corresponding DIR chart (right) showing the manually drawn polygons of different degrees of ice ridging, including the marginal ice zone detection mask based on the ice concentration values between 25 % and 80 % and open-water mask based on the ice concentration values smaller than 25 %.

Figure 2. A 20 km section of combined surface laser profile and EM thickness profile, and the corresponding ice thickness histogram for the 2011 field campain data. The laser profile resolves all ridge sails while the EM profile averages thickness over an altitude dependent footprint, typically 50 m.

Figure 3. Example of RS-2 SAR data mosaic in HH and HV polarization and the corresponding DIR chart with values extracted from the digitized Finnish ice charts, for 9[th] March 2013 in the upper panel and 12[th] March in the lower panel. In both days the SAR shows similar ice situation, albeit the two DIR charts show changes in the ridging classes: in the NW of the Bay of Bothnia, from slightly ridged ice (DIR 2) to level-ice (DIR 1) and in the central to southern part of the Bay of Bothnia, from heavily ridged ice (DIR 4) to slightly ridged ice (DIR 2). In this case, the data from 9[th] March 2013 was removed from the classification.

Figure 4. Example of RS2 SAR data from 9[th] February 2013 in HH (top left) and HV (top right) polarizations together with the segmentation result (bottom left) and the SIC chart (bottom right).

Figure 5. Example of SAR features computed for central part of the Bay of Bothnia. a-b) original SAR HH and HV in 500 m resolution; c) Segmentation result of the first principal component of the original HH and HV SAR channels; d) SIC (1-100%); e) FIS DIR (1-4); f-g) segment means; h) $AC_{HH}$; i) $AC_{HV}$; j) $E_{HH}$; k) $E_{HV}$; l) $CV_{HH}$; m) $CV_{HV}$; n) $ED_{HH}$; o) $ED_{HV}$; p) $K_{HH}$; q) $K_{HV}$.

Figure 6. Ridge density variation in the test area (upper panel, left), HEM thickness measurements (upper panel, right), DIR indices (lower panel, left), histogram of ridge densities determined for one 1x1 NM cell (lower panel, right).

Figure 7. The monthly HH-polarization backscattering coefficient distribution for level (dashed line) and ridged (solid line) ice areas. The results are for January (left), February (middle) and March (right) 2013.

Figure 8. The monthly HV-polarization backscattering coefficient distribution for level (dashed line) and ridged (solid line) ice areas. The results are for January (left), February (middle) and March (right) 2013.

Figure 9. The detection rates for the different DIR categories in all the classification.

Figure 10. Degree of ice ridging extracted from the digitized Finnish Ice Charts on 9[th] February 2013 (left) and the result of estimated DIR based on our RF approach (right). The DIR charts includes the marginal ice zones (25 %<IC<80 %) extracted from the ice concentration charts (see Fig. 4).

Figure 11. Example of RS2 SAR data on 15[th] March 2013 in HH (top left) and HV (top right) polarizations. Middle left: MRF MMD segmentation result for the HH-HV first PCA component. Middle right: ice concentration chart extracted from the Finnish ice chart.

Figure 12. Degree of ice ridging extracted from the digitized Finnish ice charts on 15th March 2013 (left); Result of estimated DIR based on our RF approach (right). The DIR charts includes the marginal ice zones (25 %<IC<80 %) extracted from the ice concentration charts (see Fig. 11).

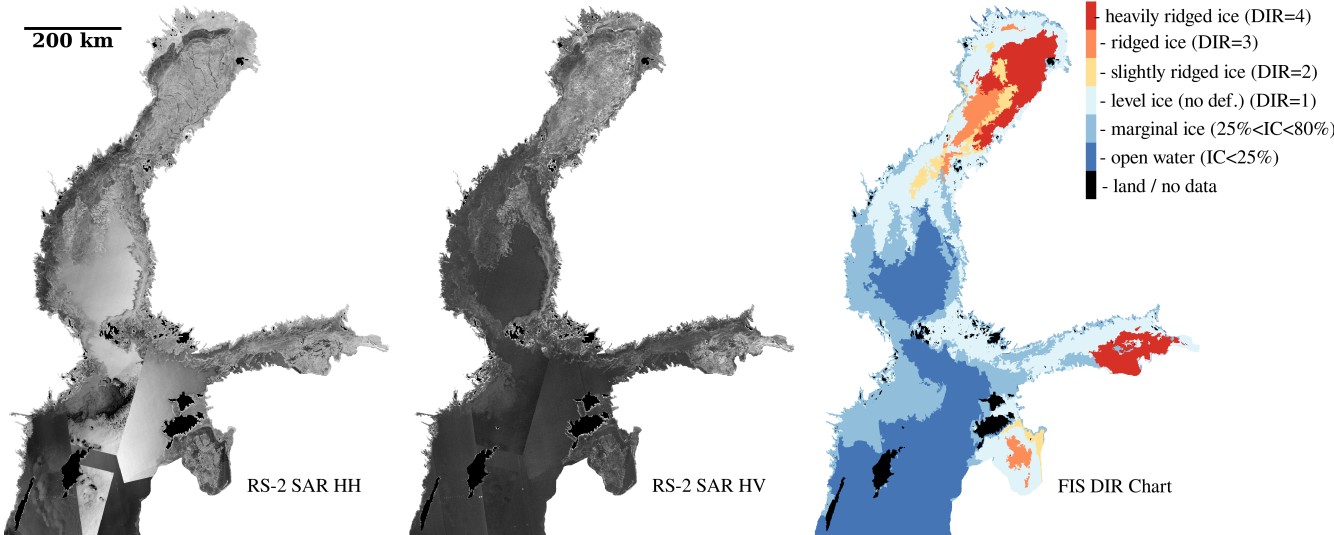

**Figure 1.** Example of RS-2 dual polarized SAR image mosaic (left: HH, middle HV) over the Baltic Sea on 15[th] March 2013 and the corresponding DIR chart (right) showing the manually drawn polygons of different degrees of ice ridging, including the marginal ice zone detection mask based on the ice concentration values between 25 % and 80 % and open-water mask based on the ice concentration values smaller than 25 %.

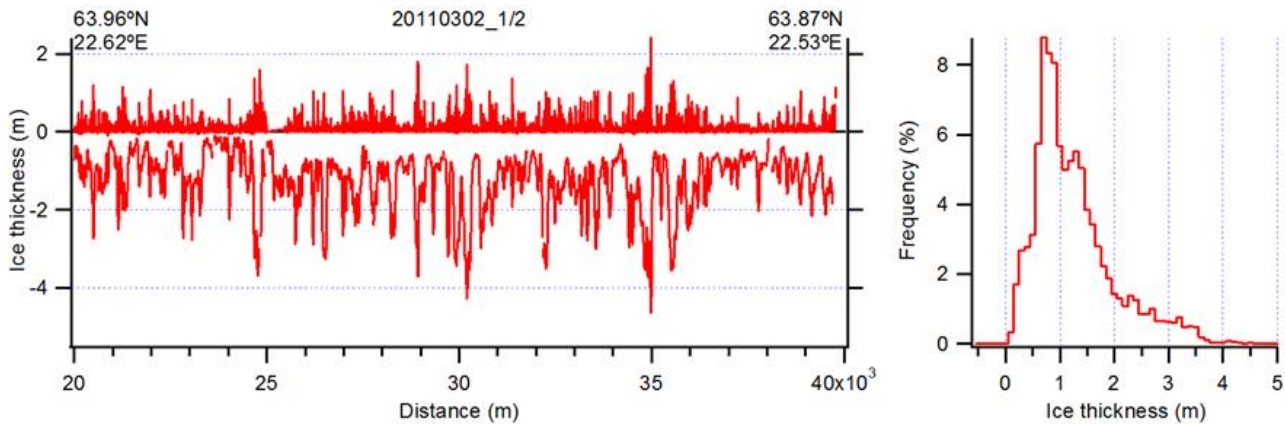

**Figure 2.** A 20 km section of combined surface laser profile and EM thickness profile, and the corresponding ice thickness histogram for the 2011 field campain data. The laser profile resolves all ridge sails while the EM profile averages thickness over an altitude dependent footprint, typically 50 m.

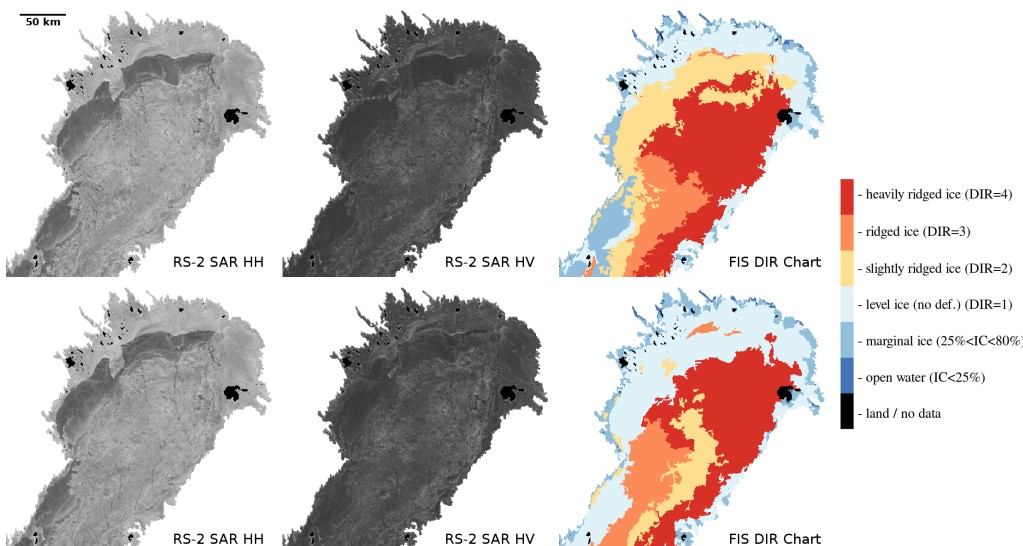

**Figure 3.** Example of RS-2 SAR data mosaic in HH and HV polarization and the corresponding DIR chart with values extracted from the digitized Finnish ice charts, for 9[th] March 2013 in the upper panel and 12[th] March in the lower panel. In both days the SAR shows similar ice situation, albeit the two DIR charts show changes in the ridging classes: in the NW of the Bay of Bothnia, from slightly ridged ice (DIR 2) to level-ice (DIR 1) and in the central to southern part of the Bay of Bothnia, from heavily ridged ice (DIR 4) to slightly ridged ice (DIR 2). In this case, the data from 9[th] March 2013 was removed from the classification.

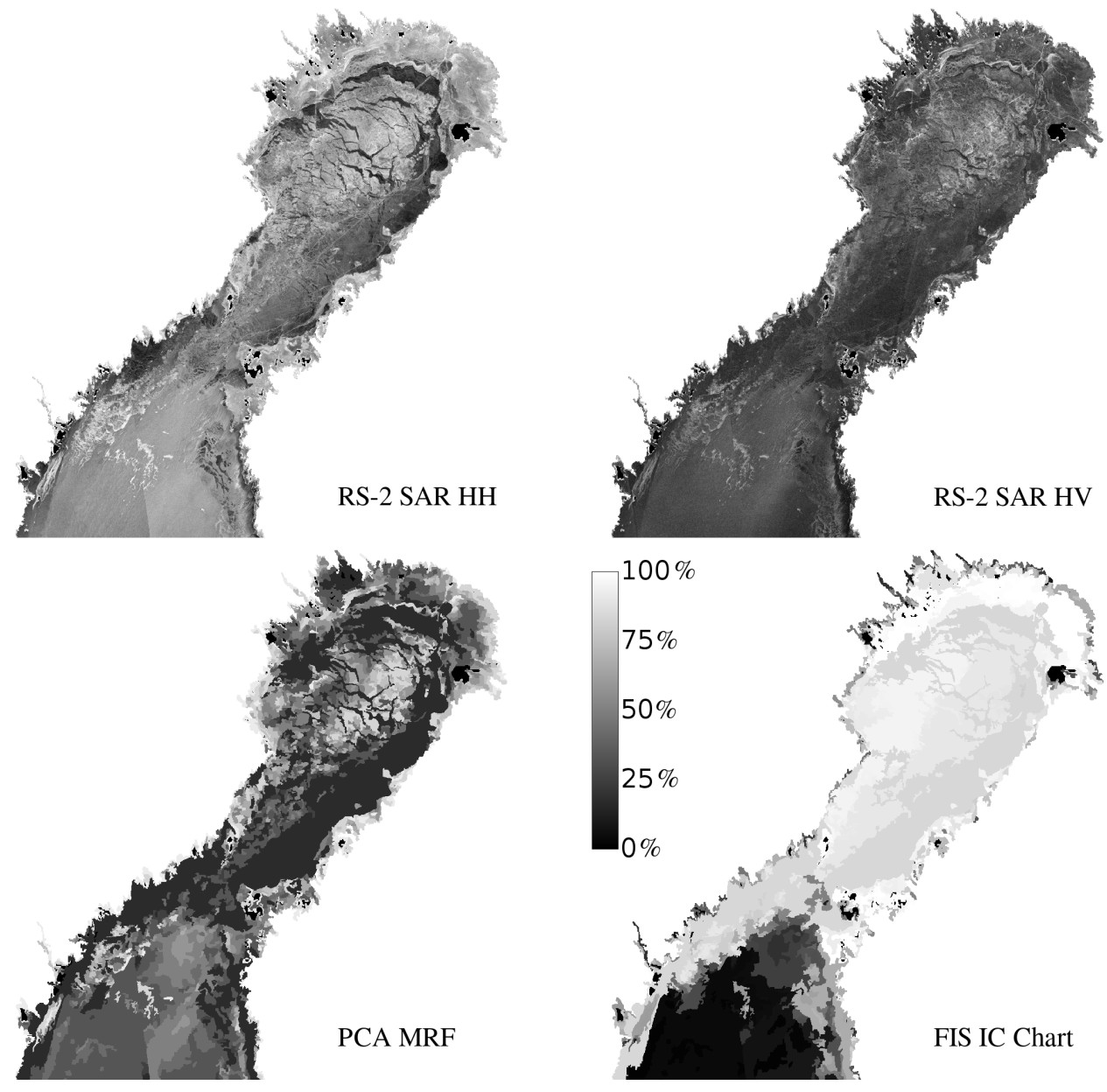

**Figure 4.** Example of RS2 SAR data from 9[th] February 2013 in HH (top left) and HV (top right) polarizations together with the segmentation result (bottom left) and the SIC chart (bottom right).

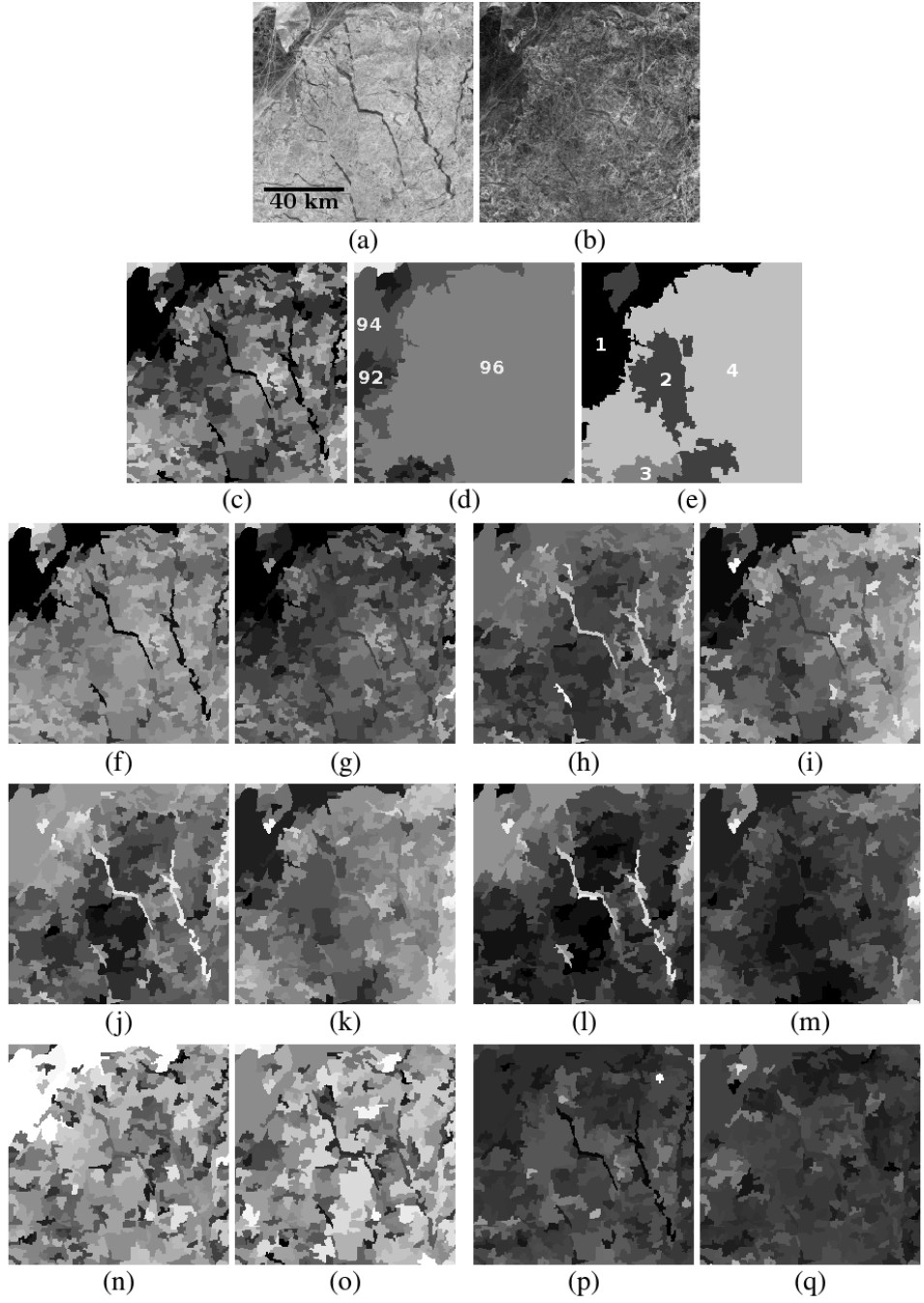

**Figure 5.** Example of SAR features computed for central part of the Bay of Bothnia. a-b) original SAR HH and HV in 500 m resolution; c) Segmentation result of the first principal component of the original HH and HV SAR channels; d) SIC (1-100%); e) FIS DIR (1-4); f-g) segment means; h) $AC_{HH}$; i) $AC_{HV}$; j) $E_{HH}$; k) $E_{HV}$; l) $CV_{HH}$; m) $CV_{HV}$; n) $ED_{HH}$; o) $ED_{HV}$; p) $K_{HH}$; q) $K_{HV}$.

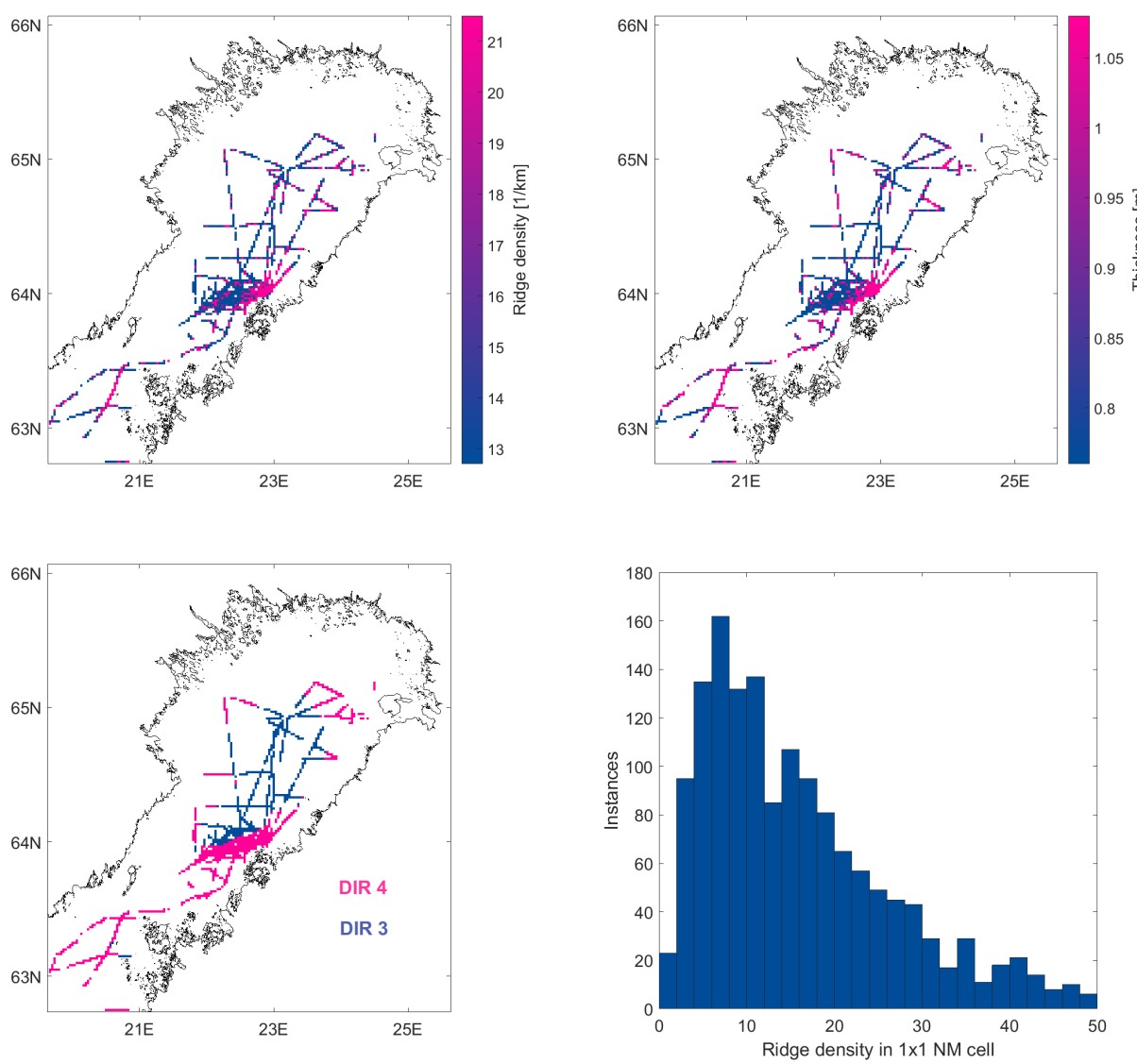

**Figure 6.** Ridge density variation in the test area (upper panel, left), HEM thickness measurements (upper panel, right), DIR indices (lower panel, left), histogram of ridge densities determined for one 1x1 NM cell (lower panel, right).

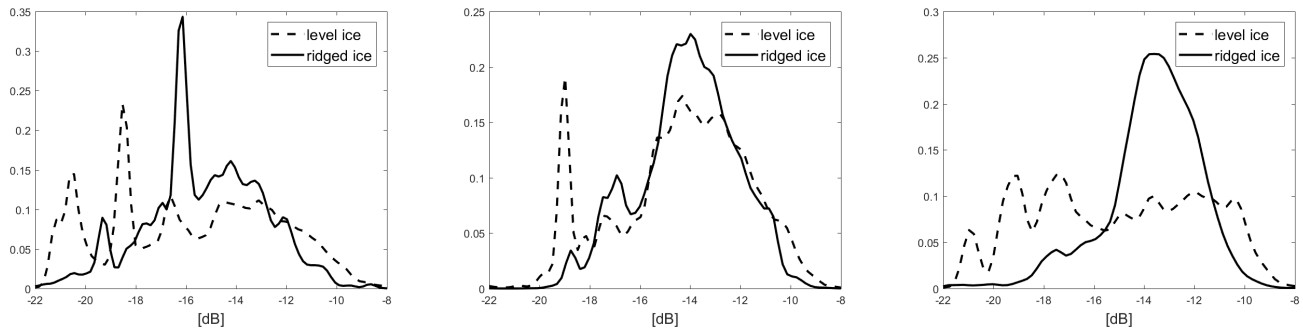

**Figure 7.** The monthly HH-polarization backscattering coefficient distribution for level (dashed line) and ridged (solid line) ice areas. The results are for January (left), February (middle) and March (right) 2013.

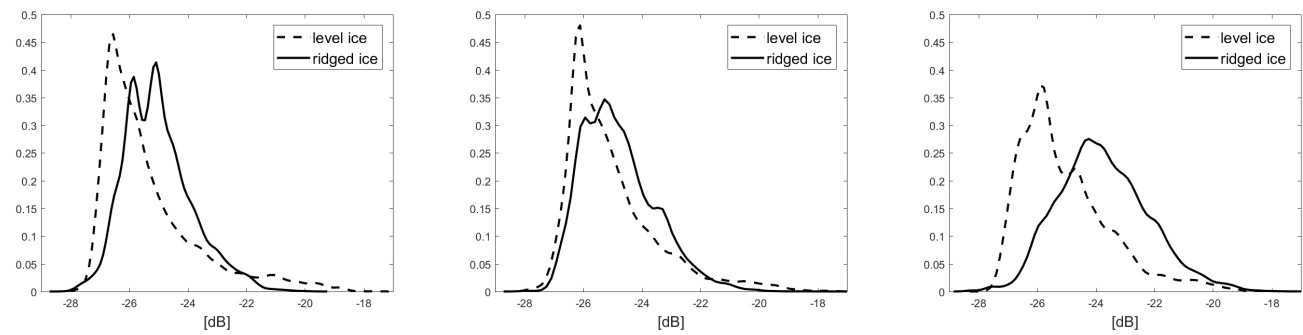

**Figure 8.** The monthly HV-polarization backscattering coefficient distribution for level (dashed line) and ridged (solid line) ice areas. The results are for January (left), February (middle) and March (right) 2013.

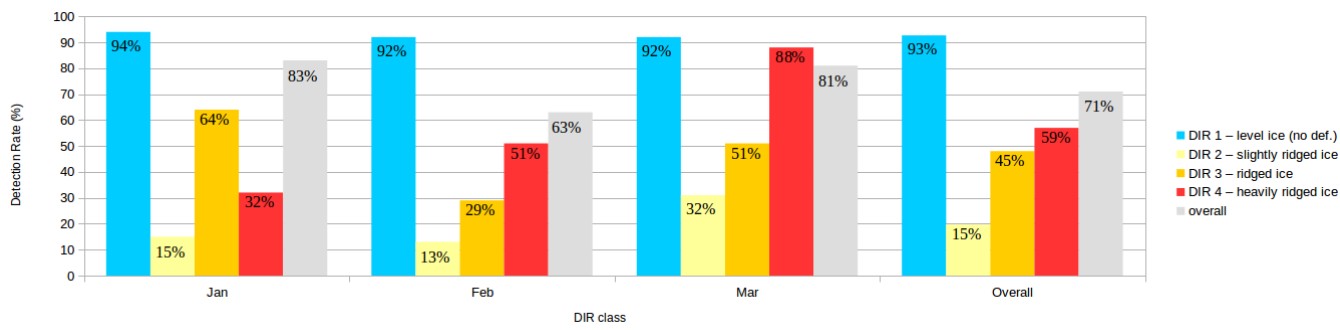

**Figure 9.** The detection rates for the different DIR categories in all the classification.

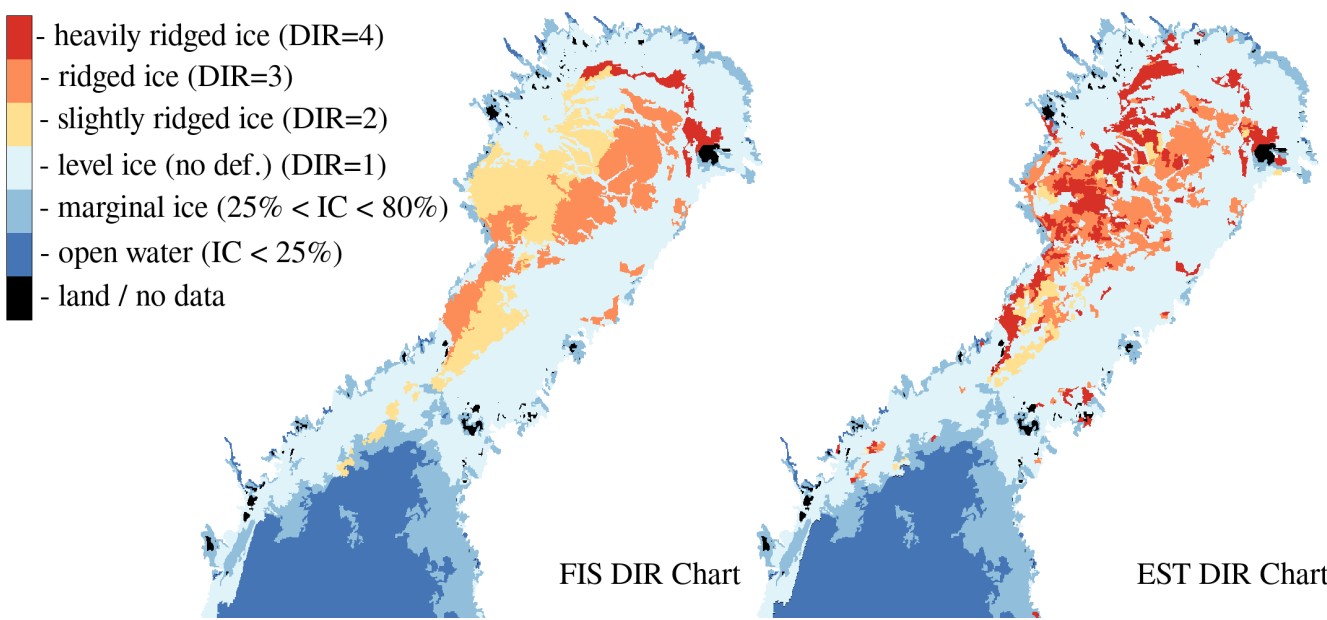

- heavily ridged ice (DIR=4)
- ridged ice (DIR=3)
- slightly ridged ice (DIR=2)
- level ice (no def.) (DIR=1)
- marginal ice (25% < IC < 80%)
- open water (IC < 25%)
- land / no data

FIS DIR Chart

EST DIR Chart

**Figure 10.** Degree of ice ridging extracted from the digitized Finnish ice charts on 9$^{th}$ February 2013 (left) and the result of estimated DIR based on our RF approach (right). The DIR charts include the marginal ice zones (25 %<IC<80 %) extracted from the ice concentration charts (see Fig. 4).

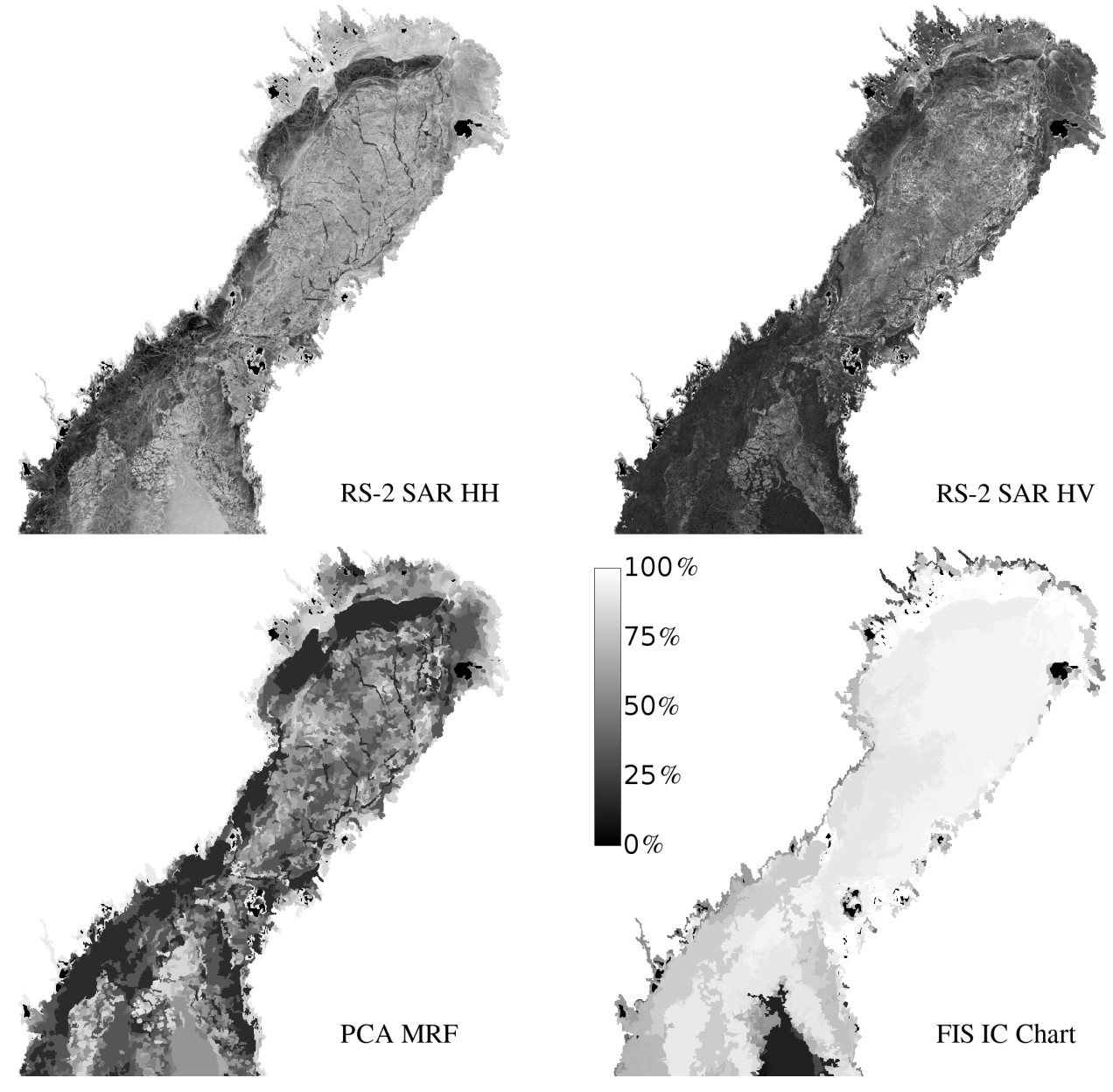

**Figure 11.** Example of RS2 SAR data on 15[th] March 2013 in HH (top left) and HV (top right) polarizations. Middle left: MRF MMD segmentation result for the HH-HV first PCA component. Middle right: ice concentration chart extracted from the Finnish ice chart.

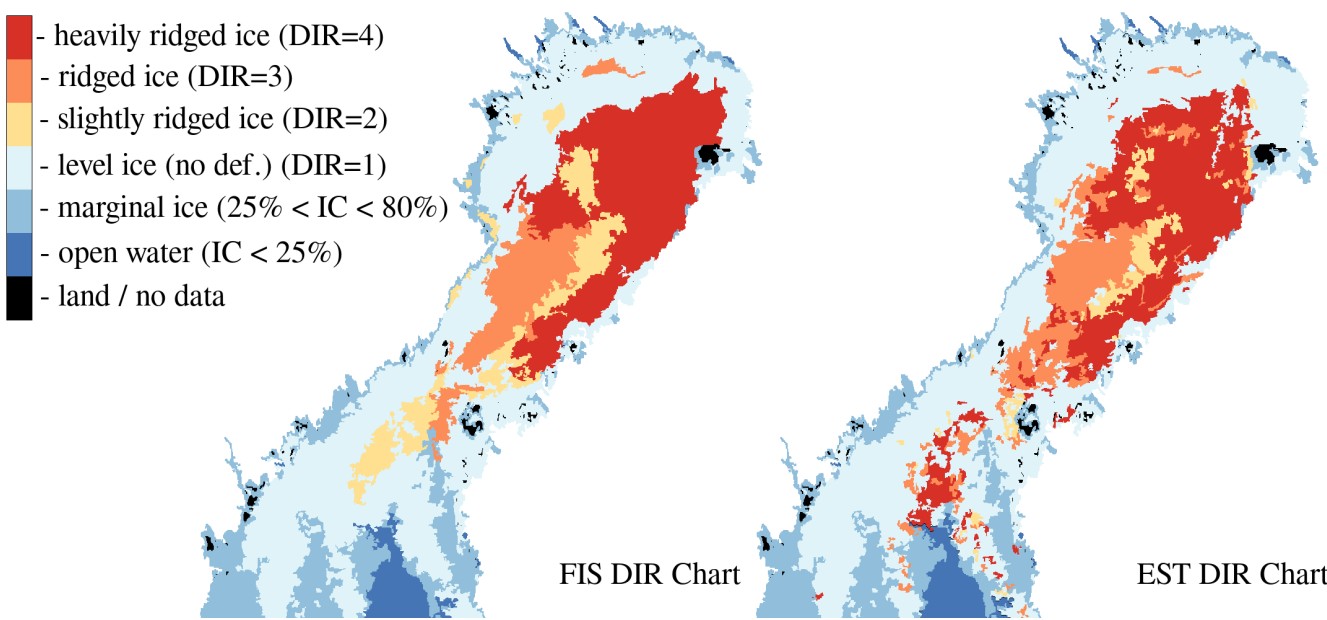

**Figure 12.** Degree of ice ridging extracted from the digitized Finnish ice charts on 15th March 2013 (left); Result of estimated DIR based on our RF approach (right). The DIR charts includes the marginal ice zones (25 %<IC<80 %) extracted from the ice concentration charts (see Fig. 11).