# Peer review of "Estimation of Degree of Sea Ice Ridging Based on Dual-Polarized C-band SAR Data"

_The Cryosphere, 2017_

## Referee Comment (RC1) · Anonymous Referee #1 · 15 Sep 2017

This is a good paper. The importance of navigation charts in ice infested seas is undeniable and ridging ice is an important parameter for navigation. The paper would benefit from a good proof reading. I put comments below that I consider should need attention: P1. Line 17: change se thickness to ice thickness P1line 21 and 23: change Seina and Peltola (1991). to (Seina and Peltola, 1991). All text: references are wrongly cited (might be LaTex-based problem) P2. L18 I wouldn't say that these egg code polygon represent uniform ice areas but uniform areas with up to 3 ice types (normally) P3. L26-34 I think this could be simplified to half of that. P4. L 8 100 m (use $\sim$ between 100 and m) P4. L 23 delete "already" P5 L 5 delete "some" P5 L 11 correct "CarlstrÂÍöm" P5 L 16 using $\sim$ between 100 and m will prevent its separation P5 L 18 use N 61° 40' P6 L 9-16 you should offer some evidence of this problem, otherwise

it seems a bit arbitrary. P7 L 11 eq. 1 should be in multiple lines... very confusing this way P12 L 4-7 This information would be better presented in a table. P12 L 11 In summary we found that the RF classification presents the following advantages: P12 section 4.1 it would greatly add value if we could see some of the field data campaign P13 L 11 the 2 top figures of figure 5 appear to be the same. P13 L 9 and L 13 first you mention Table 1 and then Table 5 is this right? P13 L 15 histogram of Figure ? P13 L 25 64N 23E to SW → 64°N 23°E to SW??? P14 L 8 values ? P 14 even though a correction for incidence angle has been applied, there is still influence of the incidence angle on the response, especially for rough ridging ice – this should also be part of the discussion on dB values: one should expect differences between near range and far range. P15 L 23 what's a " had a correct mode ice class" ? P 17 L 15-25 a bit of wishful thinking in this section Conclusions: this part appears to be more badly written than the rest. References: I saw quite a few errors including in some of the titles, authors names. One has to be careful while copying and pasting references taken "as is" on the internet; they are not always reliable.

————————————————

---

## Referee Comment (RC2) · Anonymous Referee #2 · 17 Oct 2017

Summary. This paper examines radar backscatter and texture parameters derived from Radarsat2 dual polarization imagery to determine ridge density in the Baltic Sea. A classification algorithm is described that derives several levels of ridge density, a useful parameter for shipping. Helicopter electromagnetic induction and manually derived ice charts are used for comparisons.

The paper described the concept, approach and methods adequately and was satisfactorily written, albeit some grammatical editing is needed. My primary suggestions are the following: 1) A final summary/set of conclusions of the basic findings is needed to be added. The discussion speaks in generalities about the need differences in the ice charts for both icebreakers and non-ice breakers in the Baltic plus shipping issues elsewhere in the Arctic. It seems like a key result is that the results are much better

in March than January and February. Please include a statement as to why you think this is the case. Also these differences in interpretation of the ice charts for the two types of ships seems important enough to include earlier in the paper, as it impacts final comparison results.

2) I suggest a final section be added in Section 4 that describes value of each polarization, with the HV of seemingly little to marginal value except as was pointed out perhaps in March, and the other texture parameters, in terms of what were the most important parameters in deriving ridge statistics. Could the algorithm be successful with fewer parameters? What parameters were really needed to identify ridges?? I am suggesting a further evaluation of Figure 4 basically.

Detailed comments. 1. Page 13, line 14 mentions green in Figure 5 which I assume should be red/pink. Line 15 left off Figure number, which I assume to be 5.

2. Page 13, line 25. 'to SW' I assume this means towards the SW.

3. Page 13, lines 30-31. The sentence "In areas with IC 80-90% the amount of open water is rather high...' I think they may mean that in a relatively high ice concentration area, the relatively low amount of open water can still have a strong impact on the overall backscatter, particularly during high winds. Please rewrite.

4. Page 14. Line 8. A question mark appears in text without any apparent meaning.

---

## Author Comment (AC1) · 14 Nov 2017

**Response to referee 1: Estimation of Degree of Sea Ice Ridging Based on Dual-Polarized C-band SAR Data by Alexandru Gegiuc et al.**

This is a good paper. The importance of navigation charts in ice infested seas is undeniable and ridging ice is an important parameter for navigation. The paper would benefit from a good proof reading.
**We thank the reviewer for the appreciation given to the topic discussed here. We have carefully improved the readability of the paper.**

I put comments below that I consider should need attention:
P1. Line 17: change se thickness to ice thickness
**Done.**

P1line 21 and 23: change Seina and Peltola (1991). to (Seina and Peltola, 1991). All text: references are wrongly cited (might be LaTex-based problem)
**Corrected.**

P2. L18 I wouldnt say that these egg code polygon represent uniform ice areas but uniform areas with up to 3 ice types (normally)
**In the daily ice charts provided by the Finnish Ice Service is not used the egg code for the polygons. The sentence is corrected to:**
**"The ice chart polygons defined by the ice analysts represent ice areas with similar ice characteristics."**

P3. L26-34 I think this could be simplified to half of that.
**In the context of the discussed topic of sea ice ridging and its importance for navigation, we consider that the proposed (full length) paragraph describing the Baltic Sea ice conditions for the selected data set is relevant and needed for a better understanding and interpretation of the obtained classification results. Therefore we propose to keep the paragraph unshortened.**

P4. L 8 100 m (use ∼ between 100 and m)
**Done.**

P4. L 23 delete "already"
**Done.**

P5 L 5 delete "some"
**Done.**

P5 L 11 correct "CarlstrÅlöm"
**Done.**

P5 L 16 using ∼ between 100 and m will prevent its separation
**Corrected.**

P5 L 18 use N 61° 40'
**Done.**

P6 L 9-16 you should offer some evidence of this problem, otherwise it seems a bit arbitrary.
**We agree and we have added a figure (also below as Fig. 1) with an example of an accepted and rejected SAR - chart pair in the data selection process, in the Section 2.4 (in the previous version 2.3).**

[Figure]

Figure 1: Example of RS-2 SAR data mosaic in HH (left) and HV (middle) polarizations and the corresponding degree of ice ridging chart (right) with values extracted from the digitized Finnish ice charts, for 9$^{th}$ March 2013 in the upper panel and 12$^{th}$ March in the lower panel. In both days the SAR shows similar ice situation, albeit the two DIR charts show changes in the ridging classes: in the NW of Bay of Bothnia, from slightly deformed ice (DIR=2) to level-ice (DIR=1) and in the central to southern part of Bay of Bothnia, from heavily ridged ice (DIR=4) to slightly ridged ice (DIR=2). In this case, the data from 9$^{th}$ March 2013 was removed from the classification.

**The example is discussed in detail in the Section 2.4 (in the previous version 2.3).**

P7 L 11 eq. 1 should be in multiple lines... very confusing this way
**Done.**

P12 L 4-7 This information would be better presented in a table.
**We have replaced the figure with a table (see Table 1).**

Table 1: The importance of different features when the training data covered the whole test period.

| feature | FIS SIC | $K_{HH}$ | $\sigma^0_{HH}$ | $ED_{HH}$ | $AC_{HH}$ | $E_{HH}$ | $CV_{HH}$ | $\sigma^0_{HV}$ |
|---|---|---|---|---|---|---|---|---|
| importance (%) | 13.9 | 11.9 | 11.7 | 11.3 | 8.1 | 7.2 | 7.2 | 6.9 |

P12 L 11 In summary we found that the RF classification presents the following advantages:
**We have added the following paragraph at the end of the Section 3.3.2 which summarizes advantages and disadvantages for the RF classifier in the DIR estimation.**
**"In summary we found that the RF classification presents the following advantages : i) RF has the ability to describe complex, nonlinear statistical relationships among variables, ii) RF reduces the uncertainty of the obtained estimate, iii) RF reduces the possibility of over fitting. The greatest weakness in RF is its relatively weak extrapolation property (Hastie et. al., 2011). This property can be seen from the behavior of the error rates. The RF classifier has a very low training error rate but the error rates increases significantly for the test set."**

P12 section 4.1 it would greatly add value if we could see some of the field data campaign
**Based on the Reviewer's recommendation, we have added a figure (see Fig. 2) with the sea ice field campaign data into the Section 2.3 (in the previous version 2.1) entitled now "Surface and thickness profile data on ridged ice", where this data is discussed.**

[Figure]

Figure 2: A 20 km section of combined surface laser profile and EM thickness profile, and the corresponding ice thickness histogram for the 2011 field campaign data. The laser profile resolves all ridge sails while the EM profile averages thickness over a altitude dependent footprint, typically 50 m.

P13 L 11 the 2 top figures of figure 5 appear to be the same.
**The ridge density figure and the measured thickness values figure resembles each other very much, but a careful examination of the two figures reveals some small differences between the two. The very similarity between the two results from the high correlation between the measured thickness and the computed ridge density. More details are discussed in Section 4.1.**

P13 L 9 and L 13 first you mention Table 1 and then Table 5 is this right?
**There was an error. In both cases we referenced to the Table 2 (previously Table 1).**

P13 L 15 histogram of Figure ?
**Corrected. The reference is to the Fig. 6 (previously Fig. 4).**

P13 L 25 64N 23E to SW → 64°N 23°E to SW???
**Corrected to : 64°N 23°E towards SW**

P14 L 8 values ?
**A reference was missing which generated the question mark. The sentence is now as follows: *For thin ice rather high $\sigma_{HH}^o$ (over $-18$ dB) have also been observed (Mäkynen et al., 2004).***

P 14 even though a correction for incidence angle has been applied, there is still influence of the incidence angle on the response, especially for rough ridging ice this should also be part of the discussion on dB values: one should expect differences between near range and far range.
**Based on the incidence angle correction methods studied in Mäkynen et al. (2002) and Karvonen (2002), the effect of the incidence angle on the level ice and ridged ice is minor for the $\sigma_{HH}^0$. For the $\sigma_{HV}^0$, after the incidence angle correction has been applied together with the noise floor correction in the range direction as in Karvonen (2015), there was not found any significant difference in the near-range and far-range values for either open-water, level ice or ridged ice classes, when visually inspecting the SAR images.**

**We agree that the incidence correction effect on the detection of level ice and ridged ice should be mentioned along with the backscattering statistics presented in Section 4.2. Therefore, we have added the following paragraphs in the Section 4.2:**

**"According to earlier studies the effect of incidence angle on $\sigma^o$ for level ice and ridged ice is rather similar. In Mäkynen et al. (2002) it was found that the incidence angle dependence of $\sigma_{HH}^o$ in logarithmic scale (dB) can be described by a linear model, with slopes -0.21 dB/degree for ridged ice and -0.25 dB/degree for level ice. It seems that using a slope of -0.23 for all the data is adequate for automated classification, and the ridged areas and level ice can be distinguished both at near and far range. Also a more sophisticated approach, iteratively applying different slopes for level ice and ridged ice has been studied in Karvonen (2002), but the effect on sea ice clas-**

sification was minor. When inspecting the SAR mosaics visually most of the SAR frame boundaries were not visible or were hardly visible, indicating successful $\sigma^o_{HH}$ incidence angle correction. For open water the correction may not work properly as for open water $\sigma^o$ signatures depend heavily on wind speed and swell (i.e., surface roughness).

For HV channel, the combined incidence angle and noise floor correction is essential. Without this correction the HV backscattering and texture features derived from it can not be used in classification as the effect of the varying noise floor is so high (up to about 3 dB) and will cause a significant amount of misclassifications. However, after correction the HV channel data can be used in classification and we have not by visual inspection observed any significant differences in near-range and far-range $\sigma^o_{HV}$ for either open water, level ice or ridged ice classes."

P15 L 23 whats a "had a correct mode ice class" ?
Sentence corrected to: "The ridged ice category (DIR 3) was correctly classified in 45 % of the cases but over 30 % of the observations were confused with level ice."

P 17 L 15-25 a bit of wishful thinking in this section
We are not certain which part of the text is considered by the Reviewer 1 as "wishful thinking". We admit that our current algorithm may not be mature enough for operational use yet. This will require more testing with more data.

Conclusions: this part appears to be more badly written than the rest.
Also the Reviewer 2 had some concerns on this Section. Therefore, we have rewritten it partly and tried to focus the discussion and conclusions in a more clear and focused manner. The new text is included below.

[revised manuscript text omitted]

References: I saw quite a few errors including in some of the titles, authors names. One has to be careful while copying and pasting references taken as is on the Internet; they are not always reliable.
**We have corrected the errors found in the References section; we hope that there will not be any more errors left uncorrected.**

---

## Author Comment (AC2) · 14 Nov 2017

**Response to referee 2: Estimation of Degree of Sea Ice Ridging Based on Dual-Polarized C-band SAR Data by Alexandru Gegiuc et al.**

Summary. This paper examines radar backscatter and texture parameters derived from Radarsat2 dual polarization imagery to determine ridge density in the Baltic Sea. A classification algorithm is described that derives several levels of ridge density, a useful parameter for shipping. Helicopter electromagnetic induction and manually derived ice charts are used for comparisons. The paper described the concept, approach and methods adequately and was satisfactorily written, albeit some grammatical editing is needed.
**Many parts of the text have been edited to increase the readability, including some grammatical and spelling errors.**

My primary suggestions are the following:

1) A final summary/set of conclusions of the basic findings is needed to be added. The discussion speaks in generalities about the need differences in the ice charts for both icebreakers and non-ice breakers in the Baltic plus shipping issues elsewhere in the Arctic.

**The Discussion and Conclusions section has been partly rewritten, taking into account the remarks of both the reviewers. Please, see the edited section included in our response to the Reviewer 1.**

It seems like a key result is that the results are much better in March than January and February. Please include a statement as to why you think this is the case.

**"The major reason for the success of the classification in March is the better discrimination between the ridged ice and level ice in March than in the previous months as noted earlier in Section 4.2. The better discrimination property between ridging ice categories affects the final results in two ways. First, the segment boundaries of the dual-pol SAR imagery follow better the boundaries of the DIR classes in March (see Fig. 11). Secondly, the segmentwise feature vectors show more variability between different ridging categories in midwinter. The combination of these two factors determine the accuracy of the final classification.**

**We studied the success of the segmentation by examining how large fraction of the segments contained practically just one ridging category. i.e. the area of some ridging category covered over 90 % of the segment area. The results were that in January 93 % of the SAR imagery belonged to such segments, in February 80 % and in March 86 %. The high fraction of well defined segments in January is easy to understand because most of the ice was level ice (72 % of the area), and just three ridging categories appeared (the heavily**

ridged area covered less than 1 %). In February the fraction of level ice has decreased to 55% of the total area, all four ice categories were present and the total area of well defined segments decreased to 80 %. In March the level ice area covered 59 % of the total area and the area of the well defined segments was 86 %. Hence there was better the segmentation accuracy in March than in February. In that month the total area of correctly classified ridging categories was 81 %, five percent points less than the total area of the well defined segments. In February the total area of correctly classified ridging categories was just 63 % which means 17 percent points less than the total area of the well defined segments. This analysis suggests that the main separating factor contributing to the classification accuracy was due to the more versatile feature vectors in March."

We added the above paragraphs inside the quotation marks at the end of Section 4.3.

Also these differences in interpretation of the ice charts for the two types of ships seems important enough to include earlier in the paper, as it impacts final comparison results.

We agree. We have mentioned this issue in the Introduction section as follows:

"In this paper we propose a method to automatize the DIR estimation process based on th RS-2 dual-polarized (HH/HV) SAR data acquired under cold conditions and using the FIS ice charts as reference data. The results are then evaluated together with the ice analysts. We don't expect a perfect match between the automatic chart and the manual one. The polygons in the manual charts typically suppress certain amount of variation for the small-scale features and merge them into one DIR category. Here we aim to produce a more detailed DIR chart, which follows closely the SAR texture features of sea ice ridges, edges, cracks and leads. This allows the icebreakers and the non-icebreaker vessels to benefit from it in advance route planning and optimization. Ultimately, the goal is to facilitate independent sea ice navigation of non-icebreaker ships, where a finer scale DIR map can offer more sea ice passages with lower degrees of ice ridging, instead of a large polygon which either allows or denies the navigation in a specific area."

2) I suggest a final section be added in Section 4 that describes value of each polarization, with the HV of seemingly little to marginal value except as was pointed out perhaps in March, and the other texture parameters, in terms of what were the most important parameters in deriving ridge statistics. Could the algorithm be successful with fewer parameters? What parameters were really needed to identify ridges?? I am suggesting a further evaluation of Figure 4 basically.

The first correction we made was the addition of the following paragraph at the end of the Section 3.2. to justify, partly intuitively, why we have chosen the computed features.

"Most of the features have a rather straightforward interpretation. Entropy describes how uniformly the HH/HV values are distributed.

Edge density is a measure for edge fragments present in the segment which fragments we assume to be related to ridging. Coefficient of variation (CV) describes how fast the standard deviation increases with the mean. We expect that in the ridged areas CV is larger than in the homogeneous areas. Kurtosis Kurtosis describes the peakness of the $\sigma^o$ distribution. With the aid of the spatial autocorrelation we can quantify how structured the ice field in question is in the SAR imagery. We expect that more structural elements appear in the ridged ice than in the level ice where the spatial $\sigma^o$ variation is more random."

Our procedure to compute the importance of the feature is explained in Section 3.3.2. It is based on the out-of-bag (OOB) samples of the data. We also considered how well different feature combinations classified the data. To clarify the nature of the importance of the feature based on the OOB data we added the following to the added subsection 4.4 (i.e. Importance of features) in the Results section:

"The selection the eight features in Section 3.3.2 was based on their importance value. The features consisted of six HH-polarization based segment-wise features (see Section 3.2) and the segment-wise $\sigma^o_{HV}$ as well as the $IC$ value extracted from the FIS ice chart. Their importance order when the training data covered the whole test period is presented in the Table 1. If the training data of just one month was used the importance order of features varied slightly. The importance of one specific feature is relative in the sense that it changes when the combination of the used features changes, i.e. the importance of one feature depends on which other features are included. The feature IC remained however the most influential feature in every case. This is comprehensible because when IC was between 80 % and 90 %, the ice area in question represented almost always the level ice category (DIR 1) and the corresponding feature vector was easy to classify correctly. The rather low importance value of $\sigma^o_{HV}$ is probably due to the relative narrow range of the $\sigma^o_{HV}$ values.

To gain more insight into how the eight selected features affected the classification accuracy, we studied the possibility of the feature reduction using the March data as benchmark. The March data was selected because the diversity of ridging categories was largest then (see Table 7). We eliminated systematically one by one the selected features and reclassified the March test data using the remaining features. In none of the cases the classification accuracy improved with fewer features. For several removed features ( $E_{HH}$, $AC_{HH}$, $K_{HH}$, $\sigma^o_{HV}$ ) the classification accuracy decreased with just a few percent points $(1-3 \%)$. The removal of the $ED_{HH}$ feature did not practically affect the accuracy at all. A significant misclassification rate increase was observed by the reduction of the $\sigma^o_{HH}$ (-6%), $CV_{HH]}$ (-8%) and IC (-12%). In every case the relative importance of the retained features changed. Hence the importance of the features present in Table 1 is true only in the context of this specific feature combination.

To see more clearly that the features included in the feature vector complement each other and make the classification more robust, we classified the March data using only three basic features ($f_3 =$

$(IC, \sigma^o_{HH}, \sigma^o_{HV})$). **The overall accuracy was just 64%. Then we added the feature $CV_{HH}$ to $f_3$ because $CV_{HH}$ caused a significant drop in the accuracy. The accuracy remained low, only 68 %. Our conclusion of the performed analysis is that the information provided by the whole feature set is needed for a good description of ridged ice field in the SAR imagery. If already a reduction of one feature decreases the classification accuracy, the reduction of two or more features would degrade the classification further. The only feature which is perhaps unnecessary is $ED_{HH}$. It was also the most heuristic one (see Section 3.2). Because it does not decrease the classification accuracy, we have kept it in the selected feature combination. We also experimented by replacing the HH-polarization based features with their HV-polarization counterparts. This lead in all of the studied cases to the degradation of the classification accuracy."**

**These indicates that all of the features (perhaps except $ED_{HH}$) are needed in a successful classification.**

Detailed comments.
1. Page 13, line 14 mentions green in Figure 5 which I assume should be red/pink.
**We agree. We have replaced "green" with "pink".**

Line 15 left off Figure number, which I assume to be 5.
**We agree. Corrected.**

2. Page 13, line 25. to SW I assume this means towards the SW.
**Yes, we have corrected the text.**

3. Page 13, lines 30-31. The sentence In areas with IC $80 - 90$ % the amount of open water is rather high... I think they may mean that in a relatively high ice concentration area, the relatively low amount of open water can still have a strong impact on the overall backscatter, particularly during high winds. Please rewrite.
**Corrected to the form:**
**"In areas with ice concentration varying from 80 % to 90 %, the amount of open water area can impact on the backscattering statistics significantly, particularly during high winds."**

4. Page 14. Line 8. A question mark appears in text without any apparent meaning.
**The reference was missing. Now it has been added. The sentence was corrected to:**
**For thin ice rather high $\sigma^o_{HH}$ (over -18 dB) have also been observed (Mäkynen et al., 2004).**

**References**

Mäkynen, M., Hallikainen, M.: Investigation of C- and X-band backscattering signatures of the Baltic Sea ice., Int. J. Remote Sens., 25, 2061–2086, 2004.

---

## Author Response (AR2)

**Response to the Editor: Estimation of Degree of Sea Ice Ridging Based on Dual-Polarized C-band SAR Data by Alexandru Gegiuc et al.**

Editor Decision: Publish subject to minor revisions (review by editor) (21 Nov 2017) by Jennifer Hutchings Comments to the Author:

Dear Alexander Geigiuc and co-authors,

Thank you for your response to the reviewers, which has greatly improved the paper. I have some comments that I hope you can address. These are mostly related to readability and explaining the interest of your results for the general readership of The Cryosphere.

Please have an English speaker proof read the paper. Some examples of gramatical errors are provided below. Line numbers refer to the manuscript with marked up edits.

page 1, line 3: "daily in manually prepared ice charts"
**Corrected**

page 1, line 6: "of the degree of ice ridging"
**Corrected**

page 1, line9: either "using the Random Forest classification method" or "using Random Forest classification".
**Replaced with: "using the Random Forest classification method".**

page 1, line 14: Consider rephrasing. Is the word "studied" appropriate here. Perhaps clearer to say "The correspondence between the degree of ridging in ice charts and the actual ridge density was validated with data collected during and field campaign in March 2011". I removed "extensive" because this is a word that places a judgemental value on the work, without providing context. I think it is personal choice if you want to keep "extensive".
**Replaced the sentence with: "The correspondence between the degree of ice ridging reported in the ice charts and the actual ridge density was validated with data collected during a field campaign in March 2011".**

page 2, line 2: a comma in front of "which"
**Done.**

page 2, line 11: comma needed before "the expertise"
**Done.**

page 2, first paragraph: Perhaps it would help the reader to identify why you talk about particular features, such as salinity, in the context of making ice charts. I think you only need a sentance in the paragraph explaining that the parameters you describe affect interpretation of remotely sensed data in classification of ice type.

**We have added the following sentence at the end of the mentioned paragraph: "The low salinity level affects the radar signal response from satellite imagery, resulting in more volume and less surface scattering of the incident signal." The SAR signal response from Baltic sea ice, including the effect of salinity is discussed in more detailed in Section 2.1.**

page 2, line 27: Not usual to start a sentence with "Because". This sentence needs to be rephrased also.

**We replaced the sentence with a new paragraph: "Typically, the ice situation changes little from one day to the next. Hence, when drawing a new ice chart, the ice analysts are using the latest chart as the basis for the new one, and only adjusting the polygon contours and their assigned DIR values to match the new ice situation. This procedure speeds up the process of ice charting but may also introduce a bias if old polygons are used. The quality of the displayed SAR features of sea ice (e.g. magnitude of contrast/intensity, amount of radar noise), the analyst's experience and their style of drawing (more detailed or less detailed) can contribute further to inconsistencies in the finalized ice chart."**

page 2, line 29: remove "th"

**Done.**

page 2, line 33: "certian amounts of"

**Corrected**

The last sentence of this paragraph is overly complicated. Simplify if possible.

**We have replaced the last two sentences in this paragraph, with the following text: "This would allow the icebreakers and non-icebreaker vessels to benefit from it in advance route planning and optimization, by taking advantage of the sea ice passages within ridged ice areas. Manual ice charting should also benefit from a more detailed and automated DIR map which can serve as basis layer for the final ice chart."**

page 3, line 6 & 7: Reconsider if "to" is appropriate in this sentence in two locations.

**Removed both "to" from this sentence.**

page 3, line 11: again "to" not needed here.

**Removed "to" from this sentence.**

page 3, line 10 to end of section: Perhaps a subsection is appropriate for this detailed information, in section 2.1? With line 9 being the end of the introduction. It is traditional for the paragraph at the end of the introduction to be a map for reading the paper.

**Agreed. We have moved the discussion related to the behaviour of $\sigma^o$ of Baltic sea ice in the newly formed Section 2.1.**

page 4, line 13: missing space "13 was".
**Corrected.**

page 4, line 15: "weather remained similar"
**Corrected.**

page 4, line 19: "The 15th of March ice extent reached"
**Corrected.**

page 4, line 21: consider using the word "further" rather than "any more".
**Replaced "any more" with "further".**

page 5, line 2: "means" not an appropriate word here. A period can not mean! "the test period includes the 13 SAR mosaics"
**Corrected.**

page 5, line 17: Specify that the -28db floor is nadir (is this correct?)
**The specified value (noise floor) is not and cannot be for nadir, since the SAR antenna does not measure at nadir. The specified nominal value for noise floor, is applicable for the whole incidence angle range, according to the SAR specifications.**

page 5, line 32: check italicisation of "where".
**Corrected.**

page 6, line 6 and elsewhere: remove space in front of '.'
**Corrected.**

page 6, line 7: "in the resolution" → "at the resolution"
**Corrected.**

page 6, line 15: rephrase " DIR 5 indicades brash ice ..... " I am not sure what you mean by "barrier" and while is not an appropriate conjuction to use in this sentence.
It would be very helpful to discuss here what a DIR of 2, 3 and 4 actually means. How are the catagories discriminated?
**We have replaced the text describing the DIR categories and how they are assigned in the ice charts and in our classification, in a more detailed manner. The new text is below.**
**"The six DIR categories used in the operational ice charting in the Baltic Sea, relate to the ridge density variation in an area. The categories are visually identified through changes both in the $\sigma^o$ response as well as in textural characteristics in SAR imagery. Their interpretation is validated through field measurements provided by the several operating icebreakers. The task of assigning a DIR value to each ice chart polygon is a complex process requiring a good understanding of the history of the current winter season, i.e. monitoring of changes in the pack ice zone and utilizing the continuous reports on ice conditions provided by the icebreakers. In our study, we only de-**

fine four DIR categories by combining the brash ice barriers (WMO, 2010) and the heavily ridged ice category. The brash ice barriers covered a very small fraction of the sea ice area, so that they couldn't have been treated as a distinct category for classification. The very heavily ridged ice field category was not present in our dataset. The four DIR categories used in this study are defined as follows:

The level-ice area is indicated as DIR 1 which usually looks homogenous and smooth in the SAR imagery with low $\sigma^o$ response. This category includes also slightly rafted ice. The slightly ridged ice category including heavily rafted ice areas is marked as DIR 2. The ridged and heavily ridged ice areas corresponding to the DIR 3 and 4, respectively, are recognized by the changes in surface roughness at a larger scale resulting in higher $\sigma^o$ values. As their formation depends on ice pressure, knowledge of the earlier ice and weather conditions is required. The DIR 4 category in our data set included the few occurrences of brash ice barriers.

page 6, line 26: Something is missing in front of "does not make any sense". Or just delete these words.
**Sentence removed completely.**

page 6, line 29: "also" not needed in this sentance. I am not sure what this is "also" to!
**Corrected.**

page 7, line 17: Is there any affect of water pockets in the ridge on decreasing keel depth estimates? I think there was some work by Stefan Hendriks attempting to model this phenomena. Asking you check for completeness, and to provide the most up-to-date understanding of the accuracy of ridge measurements with HEM.
**We have updated the text, so that it includes the most up-to-date information. The new text is below.**
**"The main dataset is from the March 2011 campaign with approximately 600 km of measurement lines by a helicopterborne electromagnetic (HEM) sensor which combines laser surface profiling and inductive distance measurement to the ice-water interface. The measurement system was similar to that described by Haas et al. (2009). The HEM measurements give as comprehensive understanding on ridging as is obtainable from linear profiles (see Fig. 2). The two profiles provide the total thickness, and the surface laser profile resolves ridge sails. The measurement spacing of the HEM instrument is 3–4 m while the measurement response is obtained from a footprint which is typically 50 m. Standard inversion of the EM response assumes that the ice has uniform thickness and zero conductivity under the footprint. Neither holds for the roughly triangular ridge keels as their porous lower parts are permeable to electric currents. This results into underestimation of keel depths with 50 % or even more (Pfaffhuber et al. 2012) and also to underestimation of the total volume of ice."**

page 7, line 17-18. This sentence is a little awkward. How about: "Mid-basin in the Bay of Bothnia, the level ice thickness reached 60 cm with somewhat decreased ridging compared to the average winter with similar wind conditions."
**Agreed. Replaced the old sentence with the suggested one: "In mid-basin of the Bay of Bothnia, the level ice thickness reached 60 cm with somewhat decreased ridging compared to the average winter with similar wind conditions."**

page 7, line 24: "surface profiles"
**Corrected.**

page 7, line 25: "with the Rayleigh criterion"
**Corrected.**

page 7, line 29: Might be clearer if you write out "per kilometer".
**Corrected as suggested.**

Regarding the lidar data collected in previous years: Is this data collected in similar locations and ice type classifications from year to year? Where the field campaigns at the same time of year?
**"To provide interannual variation for the HEM campaign based results of Section 4.1, we use data from the 1988 campaign and from three other campaigns in 1993, 1994 and 1997, summarized in Lensu (2003). The 1993 campaign was made in February and the others in March. They measured in total 1600 km of surface profiles. The 1988 campaign covers the whole Bay of Bothnia, the 1994 campaign covers it in the S-N direction while the 1993 and 1997 campaigns cover the NE quadrant of the basin."**
**The above paragraph is added also in Section 2.4.**

page 8, line 7: "To the densities ....". Rephrase this sentence. "The densities are mostly affected by the windiness of the season ...." What do you mean by windiness? How do you quantify this?
**We replaced the sentence with: "The densities are affected mostly by the number of days with strong winds during the earlier stages of the season, when ice is less resistant to the deformation. The threshold wind speed for the onset of deformation is usually 14–16 m/s."**

page 8, line 15: It is not easy on the reader to say "0.2 m above cutoff". Perhaps state "the average sail height, for sails above 0.4m, is 0.6m in the Bay of Bothnia."
**Replaced the sentence with: "However, as a first approximation, the average height of sails exceeding 0.4 m can be assumed to be 0.6 m in the Bay of Bothnia."**

page 9, line 9: "typically consisting of optical sensors". You do not need two "or" in this sentence.
**Corrected.**

page 9, line 11: I don't think you need to mention DIR 5 here, as it was

introduced earlier and their you can mention it is not included in the analysis.
**True. We have removed that part of sentence.**

page 9, line 11: "does not" → "do not"
**Corrected.**

page 9, line 13: Does disregarding ice charts that do not agree with coincident SAR analysis bias your study? Where you always sure that the disagreement was due to different information going into the manual analysis? Please clarify that you are only rejecting data for which you have evidence that the manual classification is wrong.
**The SAR - chart data pairs in disagreement were found on random days, thus their rejection should not create a systematic bias in the classification result. Regarding the selection of the coincident SAR - chart data, we have only rejected those pairs that contained "visually" at least one DIR class assigned differently to a similar region from other days.**

page 9, line 25: "there were still a few cases"
**Corrected.**

page 9, line 26: "These differences .... ". This sentence is awkward. Rephrase.

**Below is the new text: "In some other cases, the disagreements were clearly visible. If the DIR values assigned in the ice chart for a specific SAR texture region were consistent on muliple occasions, the exception pair was eliminated being considered inadequate for the classification."**

page 9, line 32: "over the Bay of Bothnia"
**Corrected.**

page 10, top paragraph: Can you comment on whether it was the aquistion of visual observations, for example, that improved the DIR classification in these two examples? Also, proof read this paragraph for awkward grammar.
**Only visual DIR classification was used here. The change in the visually assigned DIR values may be because of a different ice analyst interpreting the SAR image content or due to some additional messages from the icebreakers.**
**We replaced P 10 L 5-8 with the following text: "For ship navigation, the low ice ridging classes (DIR 1 and 2) do not likely pose any real concern. Therefore, if they are assigned differently in different days, the shipping is not affected much. On the other hand, for the automatic classifier this is a confusing case, that leads to a decrease in discrimination power between the two ridging categories."**

page 10, line 19-20: What is a feature vector? Either state this is described later, or briefly explain.
**Added text prior Section 3.1: "Then for each segment is computed a set of SAR texture features which are related to the ice ridging**

**information. Their definitions are given below in Section 3.1. The vector with features as its components is called a feature vector.''**

page 10, line 35: should there be an ''and'' rather than a semicolon between the citations?
**Corrected.**

page 11, line 2: ''The Markov Random Field (MRF) approach ...''
**Corrected.**

page 11, line 19: ''algorithm'', one too many ifs
**page12, line 19: Corrected.**

page 11, line 24: remove ''the'' from ''the Fig. 4''.
**page12, line 24: Corrected.**

page 11, section 3.2, first paragraph: It is unclear what ''SAR features'' are when they are first mentioned. Perhaps you can say ''Twelve features are extracted for each segment from the SAR backscatter data. These are refered to as SAR features from here on.''
**page 12, The concept of a SAR feature was defined in P 12 L 25 just prior the Section 3.2.**

page 14, line 8: ''which fragments we assume'' $\rightarrow$ ''which we assume''
**Corrected.**

page 14, line 10: Kurtosis repeated. Remove one.
**Corrected.**

page 14, line 13: ''In Fig. 5''
**Corrected.**

page 14, line 17: '' we found that'', remove ''out''
**Corrected.**

page 14, line 21: is ''also'' needed here?
**Removed ''also''.**

page 14, line 28: ''and the notations used in this algorithm''
**Corrected.**

page 15, line 26: ''prevents the same features from dominating every tree''
**Corrected.**

page 16, table at top, item 1(b): ''to the bootstrapped data'' should ''to'' be ''with''?
**Yes. Corrected.**

page 16, line 21: ''from the computed 13 features introduced in Section 3.2.''
**Corrected.**

page 17, line 34: "In summary ..."
**"In" was there, at Page 17, line 25.**

page 17, line 35: missing "and" before "iii)".
**Corrected.**

page 19, line 28: "We concentrated our analysis on the areas ..."
**Corrected.**

page 19, line 29: "... can impact the backscattering statistics ..."
**Corrected.**

page 21, line 32: " Data of all three months ..."
**Corrected.**

page 21, line 18: " In none of the months was DIR 2 successfully detected due ..."
**Corrected.**

page 21, line 22: "were usually identified except". "well found" is nonsensical.
**Corrected.**

Check that the figures are referenced in order. I think fig. 12 is discussed before fig. 11 for example.
**Fig. 11 was mentioned earlier in the context of the segmentation in P 12 L 24. The logical order of the figures is that the SAR mosaic (Fig. 11) precedes the classification result (Fig. 12).**

page 21: Can you comment on the fact that the autonomous algorithm miss-classifies ice edges from cracks and leads and ridges. Or am I missinterpreting this?
**Ice edges are identified using the sea ice concentration information. We have not noticed any particular problems in their classification. E.g. if leads and cracks are large enough, then they form segments of their own which were usually well detected. It may be that we did not understood well the point in your question.**

page 22, line 22: "... better segmentation accuracy ..."
**Corrected.**

page 23, line 29: what is "importance value". I think this statement is suffering from gramatical problems.
**The importance measure is explained in P 16 L 10-12 in Section 3.3.2. We add there a new sentence. "The value of the importance measure is called an importance value." Now the lines read: "Because an ensemble of trees was used in RF and a large amount of features were utilized, the results were hard to interpret. To analyse the impact of different features on the class estimation the importance measure was**

used. The value of the importance measure is called an importance value. This measure is implemented as follows:" The sentence in P 23 L 29 is corrected to: "The selection of the eight features in Section 3.3.2 was based on their importance value (see Section 3.3.2).

Check the order Tables are addressed in the text. I think Table 1 is first mentioned after tables 2 and 3, on page 23.
**Table 1 is relabelled as Table 3 and tables 2 and 3 as tables 1 and 2.**

page 23, line 5: remove the from "the feature reduction"
**Corrected.**

page 23, line 7: "In none of the cases did the classification accuracy improve ...."
**Corrected.**

page 23, line 8: ".... decreased by just a few ..."
**Corrected.**

page 24, line 17: "... a smaller DIR value is often assigned ..."
**Corrected.**

page 24, line 19: "This is especially true for DIR 2."
**Corrected.**

page 24, line 20: Surely it is a problem for automatically generated charts that SAR imagery might not have been provided to the FIS in time for daily charts to be produced? Rephrase the last sentence in this paragraph, it is a littel awkward.
**We have removed the last sentence from this paragraph. We consider that it is not an important addition to the discussion in this paragraph.**

page 24, line 23/24: "from the easier navagatable ones"
**Corrected.**

page 24, line 24: "... relies on SAR image ..."
**Corrected.**

Based on your finding of the incosistency in classifying DIR 2, would you say this is a useful classification band to keep in the DIR scheme?
**The inconsistency of DIR 2 (slightly ridged ice) between the automated classification and the FIS charts is usually a result of the subjective assessment of an ice analyst as indicated in P 24 L 19. Hence we do not regard the high misclassification rate for DIR 2 alarming. If DIR 2 would not exist, then only two different ridging categories would appear. An ice analyst would probably assign the current DIR 2 polygons to level ice and in doing so would confuse further the separation between level ice and ridged ice. This would affect the accuracy of the automatic classification because it would be harder to**

discriminate reliably between level and ridged ice. This information would therefore be lost. Therefore, we think that our classification benefits from the inclusion of the DIR 2 category. We also believe that the existence of the DIR 2 category is important in the Arctic application

page 24, line 31: "between ridged areas" ???
**page 25, line 31: Corrected.**

page 25, line 33: "The area of level ice always exceeded ..."
**Corrected.**

page 25, line 20: "seams" is a very subjective word to use here. It is also used twice in the same sentence.
**Rephrased: "Our approach works best in the Baltic Sea when the evolution of winter has passed the freezing phase and a significant amount of ridging has occurred."**

Can tables 3-7 be combined into a single table? Prehaps with multiple panels?
**We merged tables 4-7 into one. Table 3 has a different layout than the others, so was left unchanged.**

Figure 3 caption: degree of ice ridging is capatalised throughout the manuscript text.
**Corrected.**

The readership of The Cryosphere is more general than this articles intended audience. I believe that the analysis you have presented in useful for extracting information about surface roughness from SAR C-band imagery. As such your article may be of wider interest than the operational ice charting community. Would you be able to provide some additional context in the introduction and/or conclusion sections to demonstrate the wider applicability of your results? For example, can your methods be extended to investigate changes in surface roughness, ridging throughout the Arctic? Are there any impediments to this?

Finally, while you state that the method could be applied to the Arctic Ocean as a whole, how do you know that your methodology is not dependent on the relatively fresh and thin (therefore smaller block size in ridges) nature of the ice in the Baltic? Can the method identify weathered ridges that have survived a summer? I agree that the methodology will be useful for other ice services to impliment, in seasonal ice zones. I would like the paper to be more accessible to the general readership of the cryosphere who may be interested in variability of ridging in the seasonal ice zone.

[revised manuscript text omitted]

---

## Author Response (AR3)

**Response to the Editor: Estimation of Degree of Sea Ice Ridging Based on Dual-Polarized C-band SAR Data by Alexandru Gegiuc et al.**

Editor Decision: Publish subject to technical corrections (05 Dec 2017) by Jennifer Hutchings Comments to the Author: Dear Alexander Geigiuc and co-authors,

Thank you very much for the clarification you provided in your response.

Below are a couple of responses to specific points. I have chosen "Publish subject to technical corrections" in case you would like to clarify these two points in the manuscript.

page 5, line 17: Specify that the -28db floor is nadir (is this correct?) The specified value (noise floor) is not and cannot be for nadir, since the SAR antenna does not measure at nadir. The specified nominal value for noise floor, is applicable for the whole incidence angle range, according to the SAR specifications.

Thank you very much for the clarification. Please could you consider clarifying this in the text, to help the casual reader.

**We have added this clarification in the sentence at P5 L12 :**
**"The nominal noise floor equivalent $\sigma^o$ at both HH- and HV-polarization varies along the across-track direction as -28.5 $\pm$ 2.5 dB [...]".**

page 21: Can you comment on the fact that the autonomous algorithm miss-classifies ice edges from cracks and leads and ridges. Or am I missinterpreting this? Ice edges are identified using the sea ice concentration information. We have not noticed any particular problems in their classification. E.g. if leads and cracks are large enough, then they form segments of their own which were usually well detected. It may be that we did not understood well the point in your question.

- What I was trying to express here is a question of whether lead, or floe, edges affect the ridge interpretation. I think this might be a technicality that came to mind as I was reading the manuscript. I now understand that ice edges are much larger features that what I was considering to be an edge (the floe edge). Typically the ice edge is not the edge of regions or large leads. Perhaps consider defining what you mean by an edge in your detection. I think the first time you mention this is page 2, line 34 in the unmarked up manuscript. I am also unsure how edge pixels are classified (page 11). I think this is where my confusion arose from.

**The edges used as a feature in classification, are edges detected in the SAR imagery by an edge detection algorithm (a gradient thresholding algorithm). The edges, i.e. edge pixels in the SAR image, are first detected and then the amount of edge pixels proportional to the segment size are computed for each segment. These values are then used as features. The assumption is that the more edge pixels are detected in a segment area, the more ridged ice area is present.**

The features and classification are also computed for each segment, no pixel-wise features have been used. This should not be mixed with the concept of ice edge, which is defined as a contour of ice separating ice areas above and below an ice concentration threshold. Typically ice edge is given in a coarse scale, just separating the large ice and open water areas and excluding small-scale details, such as small cracks or leads. To avoid this confusion, we added the two following sentences in the second paragraph in the Introduction section:

[revised manuscript text omitted]